# Shuffling-Aware Optimization for Private Vector Mean Estimation

**Shun Takagi** [1]   **Seng Pei Liew** [1]

## Abstract

We study $d$-dimensional unbiased mean estimation in the single-message shuffle model, where each user sends a single privatized message and the analyzer only observes the shuffled multiset of reports. While optimal mechanisms are well understood in the local differential privacy setting, the corresponding notion of optimality after shuffling has remained largely unexplored. To address this gap, we introduce the recently proposed shuffle index and use it to formulate the post-shuffling mechanism design problem as an explicit optimization problem. We then establish a lower bound on the achievable mean squared error in terms of the shuffle index, which implies that mechanisms that are optimal under LDP can become suboptimal once shuffling is applied. Finally, we construct an asymptotically optimal mechanism in the high privacy regime, which as a consequence achieves a privacy-utility trade-off nearly identical to that of the central Gaussian mechanism.

## 1. Introduction

Differential privacy (DP) (Dwork, 2006; Dwork & Roth, 2013) enables principled privacy-preserving data analysis. In the central model, a trusted curator collects raw data and releases a privatized aggregate, achieving strong utility at the cost of a strong trust assumption. In the local model (LDP) (Duchi et al., 2013), each user randomizes their data before communication, removing the need for trust but often degrading accuracy significantly. The shuffle model (Erlingsson et al., 2019; Cheu et al., 2019; Balle et al., 2019; Feldman et al., 2022) offers an appealing intermediate point: users send locally randomized messages through an anonymization layer (the shuffler), which permutes the reports before analysis. By breaking the linkage between

users and messages, shuffling amplifies privacy and can approach central-model utility.

We focus on the single-message randomize-then-shuffle setting (Balle et al., 2019), where each user sends exactly one privatized message. This is the simplest and most practical instantiation of the shuffle model, and it also serves as a basic component for understanding more complex multi-message constructions (Asi et al., 2024). Despite its importance, optimal mechanism design for fundamental statistical tasks in the single-message shuffle model remains poorly understood.

In this work we study $d$-dimensional mean estimation over the unit $\ell_2$ ball $\mathbb{B}_2^d$. Given inputs $x_1, \ldots, x_n \in \mathbb{B}_2^d$, the goal is to estimate the mean $\frac{1}{n} \sum_{i=1}^{n} x_i$ from the shuffled multiset of privatized messages. While optimal mechanisms are well understood under LDP (Asi et al., 2022), the analogous question after shuffling has remained largely unexplored: which local randomizer yields the best accuracy under a shuffled DP constraint? A key difficulty is that privacy after shuffling is inherently global and depends on $n$ and on privacy amplification, making it hard to pose an explicit optimization problem in terms of the local randomizer alone. As a result, existing approaches often design mechanisms for LDP and then apply amplification, implicitly assuming that local optimality transfers to the shuffled setting.

We address this gap using the recently proposed shuffle index (Takagi & Liew, 2026), a scalar quantity determined solely by the structure of the local randomizer. In the high privacy regime, the privacy profile (Balle et al., 2018) of the shuffled mechanism is asymptotically governed by the shuffle index in a monotone way as $n \to \infty$, which allows us to replace a global $(\varepsilon, \delta)$ constraint by a local constraint. This yields a clean post-shuffling mechanism design problem: minimize the worst-case mean squared error (MSE) among unbiased single-message protocols subject to a shuffle-index constraint.

Our main results show that this formulation leads to a sharp theory. We prove a lower bound showing that any unbiased protocol with shuffle index $\chi$ must incur worst-case MSE at least $d\chi^2$. This bound implies that LDP-optimal mechanisms can be suboptimal after shuffling. In particular, we show that PrivUnit (Bhowmick et al., 2019), which is optimal for mean estimation under LDP, fails to attain

[1]LY Corporation. Correspondence to: Shun Takagi <shutakag@lycorp.co.jp>.

*Proceedings of the 43$^{rd}$ International Conference on Machine Learning*, Seoul, South Korea. PMLR 306, 2026. Copyright 2026 by the author(s).

the shuffle-index lower bound by a constant factor in high dimensions.

Finally, we construct an explicit mechanism that asymptotically matches the lower bound in the high privacy regime. Our blanket-mixed Gaussian local randomizer outputs a Gaussian with mean $0$ with some probability and otherwise outputs a Gaussian centered at the user's input, together with a universal unbiased estimator. We achieve MSE $d\chi^2(1 + o(1))$ as $\chi \to \infty$, establishing asymptotic optimality. Moreover, we prove a central-limit-type correspondence: in the high privacy regime $\varepsilon_n = O(\sqrt{\log n/n})$, the privacy profile of our shuffled mechanism converges to that of the central Gaussian mechanism with the matching noise level, which sharpens earlier order-level and empirical observations in (Chen et al., 2023) by establishing a precise theoretical correspondence, matching constants.

### 1.1. Related Work and Contributions

**Lower bounds.** For single-message shuffled mean estimation, most existing lower bounds are typically stated in a form that is uniform over $\varepsilon$ (Balle et al., 2019; Asi et al., 2024) (i.e., $\Omega\left(d\,n^{-2/(d+2)}\right)$), which is well suited to the moderate-privacy regime but provides limited resolution in the high privacy scaling $\varepsilon_n = o(1)$ as $n \to \infty$. In the high privacy regime, it is common to compare shuffled protocols against trivial lower bounds obtained by invoking lower bounds for mean estimation of the central model (Cai et al., 2021). However, such central-model lower bounds ignore the shuffling structure entirely and can therefore be loose. Moreover, central-model lower bounds are rarely suited to resolving fine-grained privacy–utility trade-offs (e.g., matching leading constants).

**Upper bounds.** In the single-message shuffle model, order-optimal protocols are known in the moderate-privacy regime $\varepsilon_n = \Theta(1)$ (see, e.g., Balle et al. (2019); Asi et al. (2024)). In the high privacy scaling $\varepsilon_n = o(1)$, there are also constructions achieving the corresponding order-optimal rates up to constant factors (Chen et al., 2023). Moreover, the resulting MSE is suggestive of convergence up to constants to that of the central Gaussian mechanism as $\varepsilon_n$ decreases. What has been missing, however, is a theory establishing whether and in what sense single-message shuffling can match the central Gaussian trade-off in the high privacy regime, and how the approach to the Gaussian limit depends quantitatively on the scaling of $\varepsilon_n$.

Scott et al. (2022) propose a single-message vector aggregation protocol, but their analysis incurs additional error terms stemming from coordinate/dimension sampling, which can make the resulting upper bounds loose in regimes where one aims for sharp constants. While one can alternatively formulate and optimize protocols under an explicit commu-

nication constraint, as explored, for instance, by Chen et al. (2023), in this work we primarily focus on the privacy-utility trade-off and do not attempt to optimize communication.

**Multi-message shuffling.** Multi-message shuffle protocols appear in (at least) two distinct forms. In the first, each user decomposes their contribution into multiple messages (multi-message per user), which are then shuffled jointly. These additional messages can strictly increase what is achievable: in the 1-dimensional mean estimation, there are multi-message constructions whose accuracy can surpass that of the central Gaussian mechanism at comparable privacy levels (Ghazi et al., 2021). For $d > 1$, however, no comparably efficient multi-message-per-user protocols are known that provably beat the central Gaussian mechanism.

In the second form, one repeats a single-message shuffled primitive over multiple rounds and composes the privacy losses (multi-round shuffling). This is the dominant approach in $d > 1$, where mean estimation is invoked repeatedly and one can exploit composition to convert many high privacy rounds into an overall moderate-privacy guarantee (Asi et al., 2024; Chen et al., 2023). Such multi-round constructions are known to approach the privacy-utility trade-off of the central model (Girgis et al., 2021; Erlingsson et al., 2020), which in turn highlights the importance of understanding the single-round high privacy single-message primitive sharply, since it governs the per-round error and the quality of the composed guarantee.

### Contributions.

- Using the shuffle index, we formulate post-shuffle mechanism design for unbiased single-message mean estimation as an explicit optimization problem, and prove a sharp lower bound MSE $\geq d\chi^2$ in the high privacy regime.

- We show that LDP-optimal primitives (e.g., PrivUnit) can be strictly suboptimal after shuffling and construct a blanket-mixed Gaussian mechanism that is asymptotically optimal, achieving MSE $= d\chi^2(1 + o(1))$ as $\chi \to \infty$.

- In the high privacy scaling, we establish a Gaussian-limit correspondence: the privacy-utility trade-off of our shuffled mechanism converges to that of the central Gaussian mechanism with matching noise level, and we validate the theoretical predictions via numerical experiments.

# 2. Preliminaries

## 2.1. Notations

We write $\mathbb{R}$ for the set of real numbers. For $d \geq 1$, let $\mathbb{B}_2^d := \{x \in \mathbb{R}^d : \|x\|_2 \leq 1\}$ and $\mathbb{S}^{d-1} := \{u \in \mathbb{R}^d : \|u\|_2 = 1\}$ denote the unit $\ell_2$ ball and the unit sphere, respectively. Throughout the paper, the input space is denoted by $\mathcal{X}$; we typically take $\mathcal{X} = \mathbb{B}_2^d$. Var and $\mathbb{E}$ denote variance and expectation, respectively. We use standard asymptotic notations $O(\cdot)$, $o(\cdot)$, $\Omega(\cdot)$, $\omega(\cdot)$, and $\Theta(\cdot)$.

## 2.2. Single-message Shuffling

We briefly review the single-message randomize-then-shuffle model (Balle et al., 2019).

**Definition 2.1** (Local randomizer with a blanket distribution (Balle et al., 2019)). A local randomizer is a randomized mechanism $\mathcal{R} : \mathcal{X} \to \mathcal{Y}$. For each input $x \in \mathcal{X}$, we write $\mathcal{R}_x$ for the law of the random output $\mathcal{R}(x)$. We denote by $\mathcal{R}_x(y)$ the corresponding probability density function. If there exists a scalar $\gamma \in (0, 1]$, a probability density $\mathcal{R}_{\mathrm{BG}}$ on $\mathcal{Y}$, and a family of probability densities $\{Q_x\}_{x \in \mathcal{X}}$ on $\mathcal{Y}$ such that, for every $x \in \mathcal{X}$ and almost every $y \in \mathcal{Y}$,

$$\mathcal{R}_x(y) = \gamma \mathcal{R}_{\mathrm{BG}}(y) + (1 - \gamma) Q_x(y),$$

$\mathcal{R}_{\mathrm{BG}}$ is called a blanket distribution of $\mathcal{R}$ (Takagi & Liew, 2026), and the scalar $\gamma$ is called the blanket mass.

We assume access to an ideal shuffler as a black box.

**Definition 2.2** (Single-message shuffling and shuffled mechanism). Let $\mathcal{R}$ be a local randomizer. Given a dataset $x_{1:n} \in \mathcal{X}^n$, each user $i \in [n]$ applies the local randomizer to obtain a single message $Y_i := \mathcal{R}(x_i) \in \mathcal{Y}$. The shuffler is a randomized map $\mathcal{S} : \mathcal{Y}^n \to \mathcal{Y}^n$ which samples a permutation $\pi$ uniformly at random from the symmetric group on $[n]$ and outputs $\mathcal{S}(y_1, \ldots, y_n) := (y_{\pi(1)}, \ldots, y_{\pi(n)})$. The shuffled mechanism is defined as $\mathcal{M} = \mathcal{S} \circ \mathcal{R}^n$, where $\mathcal{R}^n$ denotes the application of $\mathcal{R}$ independently to each record.

**Unbiased estimators.** Let $\mathcal{R}$ be a local randomizer. A pair $(\mathcal{R}, \mathcal{A})$ is called unbiased if

$$\mathbb{E}_{Y \sim \mathcal{R}_x}[\mathcal{A}(Y)] = x \qquad \text{for all } x \in \mathcal{X}.$$

In the dataset setting, we say that a pair $(\mathcal{R}^n, \mathcal{A})$ is unbiased if

$$\mathbb{E}[\mathcal{A}(\mathcal{S} \circ \mathcal{R}^n(x_{1:n}))] = \frac{1}{n} \sum_{i=1}^n x_i \qquad \text{for all } x_{1:n} \in \mathcal{X}^n.$$

## 2.3. Differential Privacy (DP)

We recall the notion of DP and express it in a form based on the hockey-stick divergence.

**Definition 2.3** (Hockey-stick divergence). Let $\mathcal{Y}$ be an output space. Let $P$ and $Q$ be probability distributions on $\mathcal{Y}$ that admit densities $p$ and $q$, and fix a parameter $\alpha \geq 1$. The hockey-stick divergence of $P$ from $Q$ of order $\alpha$ is

$$\mathcal{D}_\alpha(P\|Q) := \int_{\mathcal{Y}} \big[p(y) - \alpha q(y)\big]_+ \, dy,$$

where $[u]_+ := \max\{u, 0\}$.

**Definition 2.4** (DP (Dwork & Roth, 2013; Barthe & Olmedo, 2013)). A mechanism is a randomized map $\mathcal{M} : \mathcal{X}^n \to \mathcal{Z}$ that takes a dataset $x_{1:n} \in \mathcal{X}^n$ as input and outputs a random element of $\mathcal{Z}$. For each dataset $x_{1:n}$ we write $\mathcal{M}(x_{1:n})$ for the corresponding output distribution on $\mathcal{Z}$, and we denote its density by the same symbol when convenient. For a mechanism $\mathcal{M}$, $\varepsilon \geq 0$, and $\delta \in [0, 1]$, $\mathcal{M}$ is $(\varepsilon, \delta)$-DP if and only if

$$\delta_{\mathcal{M}}(\varepsilon) := \sup_{x_{1:n} \simeq x'_{1:n}} \mathcal{D}_{e^\varepsilon}\big(\mathcal{M}(x_{1:n}) \,\big\|\, \mathcal{M}(x'_{1:n})\big) \leq \delta,$$

where $x_{1:n} \simeq x'_{1:n}$ indicates that the datasets $x_{1:n}$ and $x'_{1:n}$ are *neighboring*. The function $\delta_{\mathcal{M}}(\varepsilon)$ is referred to as the privacy profile of $\mathcal{M}$ (Balle et al., 2018).

We adopt zero-out neighboring (Kairouz et al., 2021) as the adjacency relation. To this end, introduce a special symbol $\bot \notin \mathcal{X}$ and consider the extended domain $\mathcal{X}_\bot := \mathcal{X} \cup \{\bot\}$. For $a, b \in \mathcal{X}_\bot$, we write $a \simeq b$ if

$$(a = \bot \text{ and } b \in \mathcal{X}) \quad \text{or} \quad (b = \bot \text{ and } a \in \mathcal{X}).$$

Two datasets $x_{1:n}, x'_{1:n} \in \mathcal{X}_\bot^n$ are zero-out neighboring, denoted by $x_{1:n} \simeq x'_{1:n}$, if there exists $i \in [n]$ such that $x_j = x'_j$ for all $j \neq i$ and $x_i \simeq x'_i$. This is convenient for analysis in distributed settings where the dataset size is public. Moreover, the standard replace-one adjacency can be simulated by two zero-out neighboring steps. If a mechanism is $(\varepsilon, \delta)$-DP under zero-out neighboring, then it is $(2\varepsilon, 2\delta)$-DP under the replace-one adjacency. We may define the output distribution on input $\bot$ arbitrarily, and we choose it to be the blanket distribution of the local randomizer to simplify the analysis: $\mathcal{R}_\bot := \mathcal{R}_{\mathrm{BG}}$.

## 2.4. Shuffle Index

Given local randomizer $\mathcal{R}$, define the *generalized privacy amplification random variable* (Balle et al., 2019; Takagi & Liew, 2026) $\ell_\varepsilon(\cdot\,; x, x', \mathcal{R}_{\mathrm{ref}}) : \mathcal{Y} \to \mathbb{R}$ by

$$\ell_\varepsilon(Y; x, x', \mathcal{R}_{\mathrm{ref}}) := \frac{\mathcal{R}_x(Y) - e^\varepsilon \mathcal{R}_{x'}(Y)}{\mathcal{R}_{\mathrm{ref}}(Y)}.$$

Using $\ell_0(\cdot\,; x, x', \mathcal{R}_{\mathrm{ref}})$ and blanket mass $\gamma$, we introduce two quantities that summarize the privacy amplification due to shuffling.

**Definition 2.5** (Shuffle index (Takagi & Liew, 2026))**.** Let $\gamma \in (0, 1]$ be the blanket mass of $\mathcal{R}$. The *lower shuffle index* is

$$\chi_{\mathrm{lo}}(\mathcal{R}) := 1/ \sup_{x_1 \simeq x_1' \in \mathcal{X}_\perp} \sqrt{\frac{1}{\gamma} \mathrm{Var}_{Y \sim \mathcal{R}_{\mathrm{BG}}}[\ell_0(Y; x_1, x_1', \mathcal{R}_{\mathrm{BG}})]}.$$

The *upper shuffle index* is

$$\chi_{\mathrm{up}}(\mathcal{R}) := 1/ \sup_{x_1 \simeq x_1' \in \mathcal{X}_\perp} \sup_{x \in \mathcal{X}} \sqrt{\mathrm{Var}_{Y \sim \mathcal{R}_x}[\ell_0(Y; x_1, x_1', \mathcal{R}_x)]}.$$

We have the inequality $\chi_{\mathrm{up}}(\mathcal{R}) \geq \chi_{\mathrm{lo}}(\mathcal{R})$.

We consider a class $\mathfrak{R}$ of local randomizers such that every $\mathcal{R} \in \mathfrak{R}$ satisfies Assumption 2.8 of Takagi & Liew (2026) (see Section B for details). The class $\mathfrak{R}$ is substantially broader than the class of pure LDP mechanisms. In particular, it includes non-LDP mechanisms such as Gaussian-type randomizers. Roughly speaking, Assumption B.1 imposes mild regularity on the output distributions (e.g., absolute continuity for each input $x$ and the absence of extremely heavy tails), while still covering a wide range of practically relevant mechanisms. For local randomizers in $\mathfrak{R}$, the privacy profile of $\mathcal{S} \circ \mathcal{R}^n$ can be characterized (up to asymptotically negligible factors) solely in terms of the shuffle indices.

**Lemma 2.6** (Privacy profile bounds (Takagi & Liew, 2026))**.** *Let $\mathcal{R} \in \mathfrak{R}$, and set $\chi_{\mathrm{lo}} := \chi_{\mathrm{lo}}(\mathcal{R})$ and $\chi_{\mathrm{up}} := \chi_{\mathrm{up}}(\mathcal{R})$. Assume that as $n \to \infty$, $\varepsilon_n = \omega(\sqrt{1/n})$ and $\varepsilon_n = O\left(\sqrt{\log n/n}\right)$. Then, there exist sequences $e_n^{\mathrm{up}}, e_n^{\mathrm{lo}} \to 0$ as $n \to \infty$ such that*

$$f_{n,\varepsilon_n}(\chi_{\mathrm{up}}) \left(1 + e_n^{\mathrm{up}}\right) \leq \delta_{\mathcal{S} \circ \mathcal{R}^n}(\varepsilon_n) \leq f_{n,\varepsilon_n}(\chi_{\mathrm{lo}}) \left(1 + e_n^{\mathrm{lo}}\right),$$

*where*

$$f_{n,\varepsilon}(\chi) := \frac{1}{\sqrt{2\pi}\, \chi^3\, \varepsilon^2\, n^{3/2}} \exp\left(-\frac{\chi^2 \varepsilon^2 n}{2}\right).$$

*Note that $f_{n,\varepsilon_n}(\chi)$ is decreasing in $\chi$.*

Although the characterization is asymptotic, it shows that $\chi_{\mathrm{lo}}(\mathcal{R})$ controls the upper bound and $\chi_{\mathrm{up}}(\mathcal{R})$ controls the lower bound. Because $f_{n,\varepsilon}(\chi)$ decreases with $\chi$, larger shuffle indices lead to smaller $\delta$, and thus yielding stronger privacy guarantees after shuffling.

# 3. Shuffle-Index-Based Optimization for Mean Estimation

In this section, we analyze post-shuffling mechanism design for mean estimation through a formulation based on the shuffle index. We begin by stating our main result, which establishes a Gaussian-limit correspondence for the optimal

privacy-utility trade-off in the high privacy regime. To this end, we reduce the shuffled DP constraint to the shuffle index constraints. Then, we analyze the resulting optimization problem. Finally, we show that, from this perspective, PrivUnit can be strictly suboptimal after shuffling when $d > 1$.

## 3.1. Main Result

We begin by specifying the high-privacy scaling regime in which our results apply.

**Definition 3.1** ($\sigma$-high privacy regime)**.** Fix a constant $\sigma > 0$ and a sequence $\{\varepsilon_n\}_{n \geq 1}$ such that $\varepsilon_n = O\left(\sqrt{\log n/n}\right)$ and $\varepsilon_n = \omega(1/\sqrt{n})$. Let $f_{n,\varepsilon}(\sigma)$ be defined as in Lemma 2.6. We say that a sequence $\{(\varepsilon_n, \delta_n)\}_{n \geq 1}$ is in the *$\sigma$-high privacy regime* if

$$\delta_n = f_{n,\varepsilon_n}(\sigma) \left(1 + e_n\right) \qquad \text{for some sequence } e_n \to 0.$$

To build intuition for this high-privacy regime, we first recall the central Gaussian mechanism for mean estimation. For $\sigma > 0$ and $x_{1:n} \in (\mathbb{B}_2^d)^n$, define the central Gaussian mechanism $\mathsf{GM}(\sigma)$ by

$$\mathsf{GM}(\sigma)(x_{1:n}) := \frac{1}{n} \sum_{i=1}^{n} x_i + \frac{\sigma}{\sqrt{n}} Z, Z \sim \mathcal{N}(0, I_d). \quad (1)$$

The mean squared error of GM is

$$\mathbb{E}\left[\|\mathsf{GM}(\sigma)(x_{1:n}) - \bar{x}\|_2^2\right] = \frac{d\,\sigma^2}{n}. \quad (2)$$

Under the zero-out neighboring relation, the $\ell_2$-sensitivity of the mean is $1/n$. We write $\delta_{\mathsf{GM}(\sigma)}(\varepsilon)$ for the privacy profile of the Gaussian mechanism (Balle & Wang, 2018). By construction, $\mathsf{GM}(\sigma)$ satisfies $(\varepsilon_n, \delta_n)$-DP in the $\sigma$-high privacy regime.

**Proposition 3.2.** *Assume that $\varepsilon_n = O\left(\sqrt{\log n/n}\right)$ and $\varepsilon_n = \omega(1/\sqrt{n})$. Then, for each $\sigma > 0$, the sequence $\left\{\left(\varepsilon_n,\, \delta_{\mathsf{GM}(\sigma)}(\varepsilon_n)\right)\right\}_{n \geq 1}$ lies in the $\sigma$-high privacy regime.*

The proof is given in Appendix A.1. In the scaling regime $\varepsilon_n = O\left(\sqrt{\log n/n}\right)$ and $\varepsilon_n = \omega(1/\sqrt{n})$, the dominant behavior of the privacy profile is governed by the exponential term $\exp\left(-\frac{\sigma^2}{2}\varepsilon_n^2 n\right)$: since $\varepsilon_n^2 n \to \infty$ but $\varepsilon_n^2 n = O(\log n)$, $\delta_{\mathsf{GM}(\sigma)}(\varepsilon_n)$ decays between $\exp(-\omega(1))$ and $n^{-\Theta(1)}$ (and becomes polynomially small precisely when $\varepsilon_n^2 n = \Theta(\log n)$). If a mechanism $\mathcal{M}_n$ is DP with $(\varepsilon_n, \delta_n)$ in the $\sigma$-high privacy regime, then its privacy profile at level $\varepsilon_n$ is asymptotically no larger than that of the central Gaussian mechanism $\mathsf{GM}(\sigma)$; equivalently, $\mathcal{M}_n$ provides privacy that is (to leading order) at least as strong as $\mathsf{GM}(\sigma)$.

**Problem Formulation** Here, we formulate the post-shuffling mechanism design problem directly under an $(\varepsilon, \delta)$-DP constraint.

Let $\mathcal{R} : \mathbb{B}_2^d \to \mathcal{Y}$ be a local randomizer and let $\mathcal{A}_n : \mathcal{Y}^n \to \mathbb{R}^d$ be an analyzer. Given a dataset $x_{1:n} \in (\mathbb{B}_2^d)^n$, each user produces a single message $Y_i = \mathcal{R}(x_i)$, the shuffler outputs $\mathcal{S}(Y_{1:n})$, and the analyzer returns $\mathcal{A}_n(\mathcal{S}(Y_{1:n}))$. We measure performance by the worst-case squared $\ell_2$ loss

$$\mathsf{Err}_n(\mathcal{R}, \mathcal{A}_n) :=$$

$$\sup_{x_{1:n} \in (\mathbb{B}_2^d)^n} \mathbb{E}\left[ \left\| \mathcal{A}_n(\mathcal{S}(\mathcal{R}^n(x_{1:n}))) - \frac{1}{n}\sum_{i=1}^n x_i \right\|_2^2 \right]. \quad (3)$$

For $n = 1$ (i.e., $\mathsf{Err}_1$), $\mathcal{S}$ is the identity map, and $\mathcal{R}^1 = \mathcal{R}$. Let $\delta_{\mathcal{S} \circ \mathcal{R}^n}(\varepsilon)$ denote the privacy profile of the shuffled mechanism. We consider the following problem:

$$\inf_{\mathcal{R} \in \mathfrak{R}} \inf_{\{\mathcal{A}_n\}_{n \geq 1}} \limsup_{n \to \infty} n \cdot \mathsf{Err}_n(\mathcal{R}, \mathcal{A}_n) \text{ s.t.}$$

$$(\mathcal{R}^n, \mathcal{A}_n) \text{ is unbiased for all sufficiently large } n, \quad (4)$$

$$\delta_{\mathcal{S} \circ \mathcal{R}^n}(\varepsilon_n) \leq \delta_n \quad \text{for all sufficiently large } n,$$

This problem formalizes *post-shuffling* mechanism design: among all single-message shuffled protocols that are $(\varepsilon_n, \delta_n)$-DP after shuffling and unbiased for mean estimation over $\mathbb{B}_2^d$, find the one minimizing the worst-case MSE. Let $\mathsf{Err}_{\mathrm{DP}}^\star(\{(\varepsilon_n, \delta_n)\}_{n \geq 1})$ denote the optimal value for the problem (4) (i.e., $\limsup_{n \to \infty} n \mathsf{Err}_n(\mathcal{R}^\star, \mathcal{A}_n^\star)$).

**The Gaussian Limit Correspondence of Shuffling.** We now state our main result: in the $\sigma$-high privacy regime, the optimal worst-case MSE in the single-message shuffle model matches that of $\mathsf{GM}(\sigma)$ asymptotically.

**Theorem 3.3** (Gaussian Limit Correspondence). *Fix a constant $\sigma > 0$. In the $\sigma$-high privacy regime $\{(\varepsilon_n, \delta_n)\}_{n \geq 1}$*

$$d\sigma^2 \leq \mathsf{Err}_{\mathrm{DP}}^\star(\{(\varepsilon_n, \delta_n)\}_{n \geq 1}) \leq d\sigma^2 (1 + \eta(\sigma)),$$

*where $\eta(\sigma) \geq 0$ depends only on $\sigma$ and satisfies $\eta(\sigma) = O(\sigma^{-2/3})$ as $\sigma \to \infty$.*

A proof is provided in Section A.7. In other words, in the $\sigma$-high privacy regime, the optimal single-message shuffle protocol achieves a privacy-utility trade-off that is asymptotically identical to $\mathsf{GM}(\sigma)$ up to vanishing relative error. Both the lower and upper bounds are proved via a shuffle-index-based analysis developed in Section 3.2.

### 3.2. Reduction to Shuffle-Index-Based Optimization

Here, our goal is to reduce the $n$-user design problem (4) to a single-user design problem with the shuffle indices, and then to solve the resulting problem via matching lower and upper bounds.

**Single-user shuffle-index constrained problems.** We consider the following single-user design problems under shuffle-index constraints.

$$\min_{\mathcal{R} \in \mathfrak{R}, \, \widehat{x}} \quad \mathsf{Err}_1(\mathcal{R}, \widehat{x})$$
$$\text{s.t.} \quad \mathbb{E}_{Y \sim \mathcal{R}_x}[\widehat{x}(Y)] = x, \qquad \forall x \in \mathbb{B}_2^d, \quad (5)$$
$$\chi_{\mathrm{lo}}(\mathcal{R}) \geq \chi,$$

and

$$\min_{\mathcal{R} \in \mathfrak{R}, \, \widehat{x}} \quad \mathsf{Err}_1(\mathcal{R}, \widehat{x})$$
$$\text{s.t.} \quad \mathbb{E}_{Y \sim \mathcal{R}_x}[\widehat{x}(Y)] = x, \qquad \forall x \in \mathbb{B}_2^d, \quad (6)$$
$$\chi_{\mathrm{up}}(\mathcal{R}) \geq \chi.$$

Let $\mathsf{Err}_{\mathrm{lo}}^\star(\chi)$ and $\mathsf{Err}_{\mathrm{up}}^\star(\chi)$ denote the optimal values of (5) and (6), respectively.

**Relating the shuffled-DP design problem.** We now relate the $n$-user design problem (4) to the single-user problems with shuffle indices. The next lemma shows that, in the $\chi$-high privacy regime, the optimal risk under an $(\varepsilon_n, \delta_n)$ shuffled-DP constraint is sandwiched between the optimal single-user risks under the shuffle index constraint of $\chi$.

**Lemma 3.4** (Reduction to shuffle-index constraints). *Fix a constant $\chi > 0$. Let $\{(\varepsilon_n, \delta_n)\}_{n \geq 1}$ be in the $\chi$-high privacy regime. Then, given any $\eta > 0$*

$$\mathsf{Err}_{\mathrm{up}}^\star(\chi) \leq \mathsf{Err}_{\mathrm{DP}}^\star(\{(\varepsilon_n, \delta_n)\}_{n \geq 1}) \leq \mathsf{Err}_{\mathrm{lo}}^\star(\chi + \eta).$$

We perform two reductions. First, by the unbiasedness constraint, the $n$-user design problem reduces to a single-user design problem (Asi et al., 2022). Second, Lemma 2.6 converts the shuffled-DP constraint at $\{(\varepsilon_n, \delta_n)\}_{n \geq 1}$ in $\chi$-high privacy regime into shuffle-index constraints with $\chi$. The full proof is provided in Section A.2.

Consequently, we can sandwich the $n$-user design problem (4) between single-user problems under shuffle-index constraints. Thus, analyzing the lower and upper bounds of these problems yields corresponding bounds for (4).

**Lower bound.** The upper shuffle index $\chi_{\mathrm{up}}$ controls a $\chi^2$-type discrepancy of the worst-case neighboring output distributions $\mathcal{R}_{x_1}$ and $\mathcal{R}_{x_1'}$. Intuitively, larger $\chi_{\mathrm{up}}$ means that $\mathcal{R}_{x_1}$ and $\mathcal{R}_{x_1'}$ are harder to distinguish. Therefore, we can employ the Hammersley-Chapman-Robbins-type lower bound (Lehmann & Casella, 1998): under unbiasedness, small distinguishability forces a universal variance lower bound. Formally, we have the following result.

**Theorem 3.5.** *Let $\mathcal{R} \in \mathfrak{R}$ be a local randomizer with $\chi_{\mathrm{up}}(\mathcal{R}) < \infty$ and $\widehat{x}$ be any unbiased estimator. Then,*

$$\mathsf{Err}_1(\mathcal{R}, \widehat{x}) \geq d\,\chi_{\mathrm{up}}(\mathcal{R})^2.$$

The proof is found in Appendix A.3.

**Upper bound.** Next, we show that the lower bound is asymptotically attainable under a $\chi_{\mathrm{lo}}$ constraint.

**Corollary 3.6.** *As $\chi \to \infty$,*

$$\mathrm{Err}_{\mathrm{lo}}^{\star}(\chi) \ \leq \ d\,\chi^2\big(1 + \tfrac{3}{2}\chi^{-2/3} + O(\chi^{-4/3})\big).$$

Corollary 3.6 is proved constructively in Section 4, where we present an explicit local randomizer (the *blanket-mixed Gaussian* mechanism) together with a universal unbiased estimator achieving the stated risk.

### 3.3. Suboptimality of Existing Mechanisms in $d > 1$

PrivUnit (Duchi et al., 2013; Bhowmick et al., 2019) is optimal in LDP (Asi et al., 2022), but once we apply shuffling, it no longer achieves the optimal performance when $d > 1$.

**Proposition 3.7.** *Consider the input domain $\mathcal{X} = \mathbb{S}^{d-1} \subset \mathbb{R}^d$ for PrivUnit where $d > 1$. Let $\mathrm{PrivUnit}(p, \theta)$ be the PrivUnit local randomizer, and let $\chi_{\mathrm{lo}} := \chi_{\mathrm{lo}}(\mathrm{PrivUnit}(p, \theta))$. For any choice of $\{\theta_d\}_{d \geq 2}$, let $\widehat{x}_d$ be an unbiased estimator for $\mathrm{PrivUnit}(p, \theta_d)$. Then, $\chi_{\mathrm{lo}} \to \infty$,*

$$\mathrm{Err}_1(\mathrm{PrivUnit}(p, \theta_d), \widehat{x}_d) = C(\theta_d, d)d\chi_{\mathrm{lo}}^2\Big(1 + O(\chi_{\mathrm{lo}}^{-1})\Big)$$

*for some quantity $C(\theta_d, d) > 0$ depending only on $(\theta_d, d)$. Moreover, the best achievable leading constant is bounded away from 1:*

$$\liminf_{d \to \infty} \ \inf_{\theta \in [-1,1]} C(\theta, d) \ \geq \ \frac{\pi}{2}.$$

We defer the proof to Appendix A.4. Intuitively, PrivUnit is constrained to a two-level cap-vs-complement reweighting of the uniform measure on $\mathbb{S}^{d-1}$. Therefore, once the shuffle-index budget fixes the allowable reweighting magnitude, there is essentially no remaining freedom to maximize the input-direction mean shift that controls the risk, which leads to suboptimality.

**Optimality in the case of $d = 1$.** In one dimension, PrivUnit reduces to randomized response RR, which is the important primitive in shuffling (see, e.g., Balle et al. (2019); Asi et al. (2024)). RR asymptotically attains the lower bound.

**Proposition 3.8.** *Consider the input domain $\mathcal{X} = \mathbb{S}^0 \subset \mathbb{R}$ for $\mathrm{RR}(p)$. Let $\mathrm{RR}(p)$ be the randomized response local randomizer, and let $\chi_{\mathrm{lo}}$ be its lower shuffle index. Let $\widehat{x}_1$ be an unbiased estimator for $\mathrm{RR}(p)$. Then, $\chi_{\mathrm{lo}} \to \infty$,*

$$\mathrm{Err}_1(\mathrm{RR}(p), \widehat{x}_1) = \chi_{\mathrm{lo}}^2 \big(1 + \chi_{\mathrm{lo}}^{-1} + O(\chi_{\mathrm{lo}}^{-2})\big).$$

See Appendix A.5 for the proof.

*Remark* 3.9 (Kashin-based reductions). Kashin-based reductions (Chen et al., 2020; 2023; Feldman et al., 2021) provide a convenient way to turn $d$-dimensional mean estimation over $\mathbb{B}_2^d$ into a collection of scalar problems by mapping $x$ to a coefficient vector $z = z(x) \in \mathbb{R}^N$ in a fixed frame so that $x = Uz$ and $\|z\|_\infty \leq \frac{K}{\sqrt{N}}\|x\|_2$. This allows one to plug in a mechanism that is (near-)optimal in the one-dimensional setting described above at the level of the induced scalar subproblem. However, the reduction itself typically incurs a nontrivial distortion constant $K = \Theta(1)$ (and generally $K > 1$) in worst case. Consequently, the overall protocol can suffer a constant-factor loss in worst-case MSE due solely to the $\ell_2 \to \ell_\infty$ coordinate transformation.

## 4. Blanket-Mixed Gaussian Mechanism

The shuffle-index formulation implies that existing mechanisms can be suboptimal after shuffling. Here, we construct an explicit local randomizer that matches the shuffle-index lower bound asymptotically.

### 4.1. Mechanism

Algorithm 1 defines our blanket-mixed Gaussian local randomizer. Given input $x$, it outputs a blanket sample $Y \sim \mathcal{N}(0, \sigma_0^2 I_d)$ with probability $\gamma$, and with probability $1 - \gamma$ it outputs an informative sample $Y \sim \mathcal{N}(x, \sigma_0^2 I_d)$. The role of $\gamma$ is to explicitly embed a blanket into the mechanism; the mechanism is designed for amplification via shuffling.

**Unbiased estimator** Given a single message $Y \in \mathbb{R}^d$ produced by Algorithm 1, we use the linear estimator

$$\widehat{x}(Y) := \frac{1}{1 - \gamma} Y. \qquad (7)$$

This choice is universal. Indeed, for any $x \in \mathcal{X}$, Algorithm 1 outputs $Y \sim \gamma\mathcal{N}(0, \sigma_0^2 I_d) + (1 - \gamma)\mathcal{N}(x, \sigma_0^2 I_d)$, so $\mathbb{E}[Y] = (1 - \gamma)x$ and hence

$$\mathbb{E}_{Y \sim \mathcal{R}_x}\big[\widehat{x}(Y)\big] = \frac{1}{1 - \gamma}\,\mathbb{E}[Y] = x.$$

### 4.2. Analysis and Parameter Optimization

Let $\mathcal{R}^{\mathsf{BMG}}$ be the Blanket-Mixed Gaussian Mechanism (BMG) in Algorithm 1 with parameters $(\gamma, \sigma_0)$. Then its lower shuffle index admits the exact expression

$$\chi_{\mathrm{lo}}\big(\mathcal{R}^{\mathsf{BMG}}\big)^2 = \frac{\gamma}{(1 - \gamma)^2} \cdot \frac{1}{e^{1/\sigma_0^2} - 1}.$$

We can express $\sigma_0$ in terms of $\gamma$ and $\chi_{\mathrm{lo}}$. Therefore, for a fixed lower shuffle index budget $\chi > 0$, substituting $\sigma_0$ into

**Algorithm 1** Blanket-Mixed Gaussian Mechanism

1: **Input:** $w \in \mathcal{X}_\perp$, where $\mathcal{X} := \{x \in \mathbb{R}^d : \|x\|_2 \leq 1\}$ and $\mathcal{X}_\perp := \mathcal{X} \cup \{\perp\}$
2: **Parameters:** dimension $d \geq 1$, blanket mass $\gamma \in (0, 1]$, Gaussian scale $\sigma_0 > 0$
3: **Output:** a single message $Y \in \mathbb{R}^d$
4: **if** $w = \perp$ **then**
5:    Sample $Y \sim \mathcal{N}(0, \sigma_0^2 I_d)$
6:    **return** $Y$
7: **end if**
8: Sample $B \sim \text{Bernoulli}(\gamma)$
9: **if** $B = 1$ **then**
10:    Sample $Y \sim \mathcal{N}(0, \sigma_0^2 I_d)$
11: **else**
12:    Sample $Y \sim \mathcal{N}(w, \sigma_0^2 I_d)$
13: **end if**
14: **return** $Y$

the expression of the risk yields

$$\text{Err}_1 = \frac{d}{(1-\gamma)^2 \log\left(1 + \frac{\gamma}{(1-\gamma)^2 \chi^2}\right)} + \frac{\gamma}{1-\gamma}.$$

Therefore, we can easily optimize $\gamma$ numerically to minimize $\text{Err}_1$ under the constraint $\chi_{\text{lo}}(\mathcal{R}^{\text{BMG}}) \geq \chi$. This optimization yields the following result.

**Theorem 4.1.** *Fix $d \geq 1$. There exists a choice of parameters $(\gamma, \sigma_0)$ such that $\chi_{\text{lo}}(\mathcal{R}^{\text{BMG}}) \geq \chi$ and, as $\chi \to \infty$,*

$$\text{Err}_1(\mathcal{R}^{\text{BMG}}, \widehat{x}) = d\chi^2 \left(1 + \frac{3}{2}\chi^{-2/3} + O\left(\chi^{-4/3}\right)\right).$$

The detailed derivation of $\chi_{\text{lo}}$ and $\text{Err}_1$ and the proof are deferred to Appendix A.6.

### 4.3. The Gaussian local randomizer ($\gamma = 0$ case)

When $\chi_{\text{up}} > \chi_{\text{lo}}$, this upper bound can be loose, and in fact $\chi_{\text{up}} > \chi_{\text{lo}}$ holds for most local randomizers, including BMG. While the relative gap can vanish in the high-privacy regime (i.e., $\chi_{\text{up}}/\chi_{\text{lo}} \to 1$ as $\chi_{\text{lo}} \to \infty$), the indices typically remain distinct for finite parameters, so a nonzero absolute gap may persist.

A particularly interesting special case is the Gaussian local randomizer obtained by setting $\gamma = 0$ in BMG.

**Proposition 4.2.** *Consider the Gaussian local randomizer $\mathcal{R}^{\text{GL}}_x = \mathcal{N}(x, \sigma_0^2 I_d), x \in \mathbb{B}_2^d$, together with the unbiased estimator $\widehat{x}(Y) = Y$. Let $\chi_{\text{chua}} := 1/\sqrt{\text{Var}_{Y \sim \mathcal{R}^{\text{GL}}_{-e}}\left[\ell_0\left(Y; e, 0, \mathcal{R}^{\text{GL}}_{-e}\right)\right]}$ where $e \in \mathbb{S}^{d-1}$. Then, as $\sigma_0 \to \infty$ (equivalently $\chi_{\text{chua}} \to \infty$),*

$$\text{Err}_1(\mathcal{R}^{\text{GL}}, \widehat{x}) = d\chi_{\text{chua}}^2 \left(1 + \frac{9}{2}\chi_{\text{chua}}^{-2} + O(\chi_{\text{chua}}^{-4})\right).$$

See Appendix A.8 for the proof. Proposition 4.2 shows that the Gaussian local randomizer has a particularly sharp utility with respect to $\chi_{\text{chua}}$: the excess over the leading term decays at rate $O(\chi_{\text{chua}}^{-2})$, which is faster than BMG, whose excess term is $O(\chi_{\text{lo}}^{-2/3})$.

$\chi_{\text{chua}}$ is induced from the conjecture 3.2 of Chua et al. (2024), which implicates that, for $\mathcal{R}^{\text{GL}}$, the privacy profile can be upper bounded in terms of $\chi_{\text{chua}}$, namely as $n \to \infty$,

$$\delta_{\mathcal{S} \circ \mathcal{R}^{\text{GL}n}}(\varepsilon_n) \leq f_{n,\varepsilon_n}(\chi_{\text{chua}})(1 + o(1)).$$

This means that we can use $\chi_{\text{chua}}$ instead of $\chi_{\text{lo}}(\mathcal{R}^{\text{GL}})$ to upper bound the privacy guarantee. Unfortunately, the conjecture is currently open, so we cannot yet turn this observation into an unconditional privacy–utility guarantee. While our numerical search did not reveal counterexamples, this does not constitute evidence of the conjecture's validity. Thus, proving or refuting this conjecture remains a significant and challenging open problem, and would directly sharpen the privacy-utility trade-off.

## 5. Numerical Evaluation

We provide a numerical illustration of the Gaussian correspondence behavior induced by shuffling (Theorem 3.3). We plot the theoretical privacy-utility trade-offs implied by our analysis, and compare the proposed shuffled BMG mechanism (BMG) against the shuffled PrivUnit and the central Gaussian mechanism (GM) with sensitivity $1/n$.

**Setup and parameter tuning.** We consider $d$-dimensional mean estimation, but since both GM and BMG have $\text{Err}_n$ error that scales as $d$, we report the per-coordinate error RMSE $= \sqrt{\text{Err}_n/d}$. The BMG parameters $(\gamma, \sigma_0)$ are chosen using the optimization described in Section 4.2. To evaluate the $(\varepsilon, \delta)$-DP guarantee, we compute certified numerical upper bounds using the FFT-based accountant of Takagi & Liew (2026) (see Appendix C for details).

**Utility curves at fixed $(n, \delta)$ (Figure 1).** We first fix $n = 10^4, \delta = 10^{-5}$, and plot the resulting RMSE as a function of $\varepsilon$. Figure 1 shows that in the high privacy regime (i.e., small $\varepsilon$), the RMSE curve of BMG approaches that of GM. This numerically supports the prediction that shuffling yields a central-limit effect, leading to a privacy-utility trade-off that is nearly GM in the high privacy regime.

**Privacy profiles at fixed RMSE (Figure 2).** Next, we fix $n = 10^3$ and the target accuracy level to RMSE $= 3.16$, and plot the upper bounds of privacy profile $\delta(\varepsilon)$ as a function of $\varepsilon$. As shown in Figure 2, the numerically evaluated privacy profile of the shuffled BMG closely matches that of GM in the practical range of $\varepsilon$.

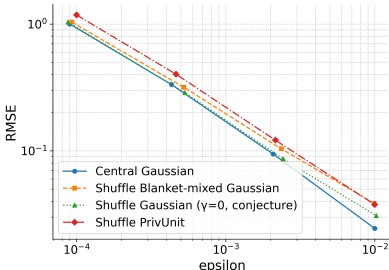 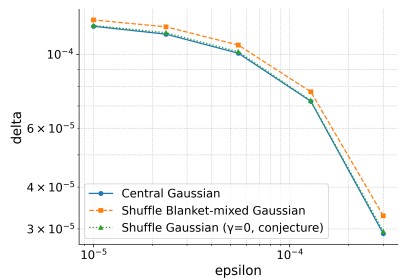 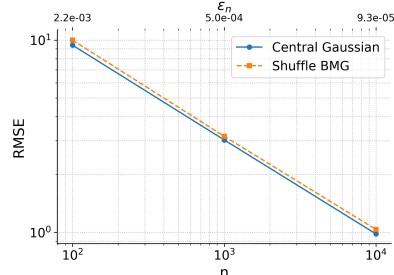

*Figure 1.* Utility comparison at fixed $(n, \delta) = (10^4, 10^{-5})$.

*Figure 2.* Privacy profile (upper bound) comparison at fixed $(n, \mathrm{RMSE}) = (10^3, 3.16)$.

*Figure 3.* Scaling with $n$ with fixed local randomizer with $(\gamma, \sigma_0) = (0.95, 4.6)$.

**Scaling with the number of users (Figure 3).** We next examine how the correspondence manifests as we vary the number of users $n$. We fix the local randomizer to a single instance of BMG same as that of Figure 2 (i.e., calibrated at $n = 10^3$ so that the shuffled protocol attains target accuracy RMSE $= 3.16$ at $\delta = 10^{-5}$). We then sweep $n \in \{10^2, 10^3, 10^4\}$, and for each $n$ compute the privacy level $\varepsilon_n$ such that BMG is $(\varepsilon_n, 10^{-5})$-DP. To compare against the GM, we calibrate the noise standard deviation so that it satisfies the same privacy constraint $(\varepsilon_n, 10^{-5})$. The resulting curves show that the RMSE of BMG closely tracks that of GM. Notably, the agreement is already strong at $n = 10^2$, indicating that the correspondence behavior induced by shuffling becomes practically relevant at small sample sizes.

## 6. Discussion

**On the unbiasedness constraint.** We impose unbiasedness throughout in order to pursue sharp constant-level optimality rather than merely order-optimal rates, in the same spirit as Asi et al. (2022). Under this restriction, we are able to (i) reduce post-shuffling mechanism design to an explicit single-user optimization problem governed by the shuffle index and (ii) prove a universal lower bound that reveals that existing mechanisms can become strictly suboptimal after shuffling. That said, unbiasedness is a substantive structural constraint, and it remains an open question whether allowing biased estimation can strictly improve the privacy-utility trade-off, potentially yielding protocols whose achievable trade-off is strictly better than the central Gaussian mechanism.

**On the high privacy regime and composition** Gaussian DP (GDP) (Dong et al., 2022) offers a clean view of multi-round composition for mean estimation with the central Gaussian mechanism. Consider running $\mathrm{GM}(\sqrt{T}\sigma)$ independently for $T$ rounds on the same dataset and releasing the average of the $T$ outputs. Because GDP is closed under composition and the squared GDP parameter adds across rounds,

this averaged $T$-round procedure has the same overall GDP guarantee as a single execution of $\mathrm{GM}(\sigma)$, and it also attains the same mean-squared error. Since single-message shuffling has a privacy profile approaching that of the central Gaussian mechanism in the high privacy regime, we expect a similar near-lossless composition behavior for multi-round shuffled protocols: repeating a high privacy single-message shuffled primitive should yield, in the moderate-privacy regime, a privacy-utility trade-off close to the central Gaussian baseline. A fully rigorous statement would require tighter composition tools, which we leave for future work.

## 7. Conclusion

In this work, we formulated and analyzed post-shuffling mechanism design as an explicit optimization problem via the shuffle index. In the $\chi$-high privacy regime, we established a sharp lower bound $d\chi^2$ on the worst-case MSE for unbiased protocols, implying that existing mechanisms can become strictly suboptimal after shuffling. We further constructed BMG that asymptotically achieves this lower bound, and proved a Gaussian-limit correspondence showing that its privacy profile converges to that of the central Gaussian mechanism under the matching noise level. This provides a principled explanation of previously observed Gaussian-like behavior under shuffling (Chen et al., 2023), sharpening earlier order-level and empirical comparisons by establishing a constant-accurate theoretical correspondence.

Several directions remain open. An important next step is to improve the convergence rate (i.e., tighten the vanishing excess term) $O(\chi^{-2/3})$. It is also of interest to extend the shuffle-index-based formulation beyond mean estimation to other fundamental estimation and learning tasks. Finally, developing tighter composition analyses for multi-round shuffled protocols may extend the constant-level correspondence beyond the high privacy regime.

## Impact Statement

This paper studies unbiased high-dimensional mean estimation in the single-message shuffle model and proposes an asymptotically optimal mechanism in the high privacy regime. Our results can contribute to the design and analysis of privacy-preserving aggregation protocols, potentially enabling safer collection of sensitive numerical data in applications such as telemetry, federated analytics, and statistical reporting.

While the primary goal of this work is methodological, improved privacy guarantees may reduce risks of data misuse and help practitioners deploy data-driven systems with stronger protections. As with many privacy technologies, however, there is a possibility that such methods could be used to justify broader data collection; we emphasize that differential privacy should be accompanied by appropriate data minimization and governance practices. Overall, we believe the expected impacts of this work are predominantly positive, supporting more trustworthy deployment of machine learning and statistical systems.

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

# A. Missing Proofs

## A.1. Proposition 3.2

**Proposition 3.2.** *Assume that $\varepsilon_n = O\left(\sqrt{\log n/n}\right)$ and $\varepsilon_n = \omega(1/\sqrt{n})$. Then, for each $\sigma > 0$, the sequence $\left\{\left(\varepsilon_n, \delta_{\mathsf{GM}(\sigma)}(\varepsilon_n)\right)\right\}_{n\geq 1}$ lies in the $\sigma$-high privacy regime.*

*Proof.* We begin by relating the privacy profile of the central Gaussian mechanism to the asymptotic template $f_{n,\varepsilon}(\cdot)$.

**Lemma A.1** (Asymptotic Gaussian privacy profile matches $f_{n,\varepsilon}(\cdot)$). *Fix any sequence $\{\varepsilon_n\}_{n\geq 1}$ with $\varepsilon_n = O\left(\sqrt{\frac{\log n}{n}}\right)$ and $\varepsilon_n = \omega\left(\frac{1}{\sqrt{n}}\right)$ and $\sigma > 0$. Then, as $n \to \infty$,*

$$\delta_{\mathsf{GM}(\sigma)}(\varepsilon_n) = f_{n,\varepsilon_n}(\sigma)\left(1 + o(1)\right).$$

*Proof.* For the Gaussian mechanism with $\ell_2$-sensitivity $\Delta$ and noise standard deviation $\sigma_{\mathrm{gm}}$, the analytic privacy profile admits the closed form (see, e.g., Balle & Wang (2018))

$$\delta_{\mathrm{GM}}(\varepsilon) = \Phi\left(\frac{\Delta}{2\sigma_{\mathrm{gm}}} - \frac{\varepsilon\sigma_{\mathrm{gm}}}{\Delta}\right) - e^{\varepsilon}\,\Phi\left(-\frac{\Delta}{2\sigma_{\mathrm{gm}}} - \frac{\varepsilon\sigma_{\mathrm{gm}}}{\Delta}\right), \tag{8}$$

where $\Phi$ is the standard normal CDF. In our setting, $\sigma_{\mathrm{gm}} = \sigma/\sqrt{n}$ and $\Delta = 1/n$. Define

$$a := \varepsilon\sigma\sqrt{n}, \qquad b := \frac{1}{2\sigma\sqrt{n}}.$$

Substituting into (8) and using $\bar{\Phi}(t) := 1 - \Phi(t)$ gives

$$\delta_{\mathsf{GM}(\sigma)}(\varepsilon) = \bar{\Phi}(a - b) - e^{\varepsilon}\bar{\Phi}(a + b). \tag{9}$$

We now use the Mills ratio expansion: as $t \to \infty$,

$$\bar{\Phi}(t) = \phi(t)\left(\frac{1}{t} - \frac{1}{t^3} + O(t^{-5})\right), \qquad \phi(t) := \frac{1}{\sqrt{2\pi}}e^{-t^2/2}. \tag{10}$$

Apply (10) with $t = a \pm b$. Since

$$\phi(a \pm b) = \phi(a)\exp\left(\mp ab - \frac{b^2}{2}\right), \qquad ab = \varepsilon/2,$$

we obtain

$$\delta_{\mathsf{GM}(\sigma)}(\varepsilon) = \phi(a)e^{\varepsilon/2}e^{-b^2/2}\left[\left(\frac{1}{a-b} - \frac{1}{(a-b)^3} + O(a^{-5})\right) - \left(\frac{1}{a+b} - \frac{1}{(a+b)^3} + O(a^{-5})\right)\right]. \tag{11}$$

Next, expand the bracketed term in $b/a$ (with $b = o(a)$, since $a \to \infty$ and $b \to 0$):

$$\left(\frac{1}{a-b} - \frac{1}{(a-b)^3}\right) - \left(\frac{1}{a+b} - \frac{1}{(a+b)^3}\right) = \frac{2b}{a^2} + O\left(\frac{b}{a^4}\right). \tag{12}$$

Substituting (12) into (11) and using $e^{-b^2/2} = 1 + o(1)$ yields

$$\delta_{\mathsf{GM}(\sigma)}(\varepsilon) = \phi(a)e^{\varepsilon/2}\left(\frac{2b}{a^2}\right)\left(1 + o(1)\right). \tag{13}$$

Finally, since $\varepsilon \to 0$ we have $e^{\varepsilon/2} = 1 + o(1)$, and plugging in $a = \varepsilon\sigma\sqrt{n}$ and $2b = 1/(\sigma\sqrt{n})$ gives

$$\delta_{\mathsf{GM}(\sigma)}(\varepsilon) = \frac{1}{\sqrt{2\pi}}\exp\left(-\frac{\varepsilon^2\sigma^2 n}{2}\right) \cdot \frac{1}{\sigma^3\varepsilon^2 n^{3/2}}\left(1 + o(1)\right),$$

which is exactly $f_{n,\varepsilon}(\sigma)\left(1 + o(1)\right)$. Applying this with $(\varepsilon, \sigma) = (\varepsilon_n, \sigma)$ completes the proof. $\square$

Let $\{\varepsilon_n\}_{n\geq 1}$ satisfy $\varepsilon_n = O\left(\sqrt{\frac{\log n}{n}}\right)$ and $\varepsilon_n = \omega\left(\frac{1}{\sqrt{n}}\right)$ and fix any $\sigma > 0$. By Lemma A.1, as $n \to \infty$ we have

$$\delta_{\mathsf{GM}(\sigma)}(\varepsilon_n) = f_{n,\varepsilon_n}(\sigma)\left(1 + o(1)\right).$$

Define $e_n := o(1)$ so that $e_n \to 0$ and

$$\delta_{\mathsf{GM}(\sigma)}(\varepsilon_n) = f_{n,\varepsilon_n}(\sigma)\left(1 + e_n\right).$$

Therefore, by Definition 3.1 (with $\chi = \sigma$), the sequence $\{(\varepsilon_n, \delta_{\mathsf{GM}(\sigma)}(\varepsilon_n))\}_{n\geq 1}$ lies in the $\sigma$-high privacy regime if $\varepsilon_n = O\left(\sqrt{\frac{\log n}{n}}\right)$ and $\varepsilon_n = \omega\left(\frac{1}{\sqrt{n}}\right)$. $\square$

## A.2. Theorem 3.4

**Lemma 3.4** (Reduction to shuffle-index constraints). *Fix a constant $\chi > 0$. Let $\{(\varepsilon_n, \delta_n)\}_{n\geq 1}$ be in the $\chi$-high privacy regime. Then, given any $\eta > 0$*

$$\mathsf{Err}^\star_{\mathrm{up}}(\chi) \leq \mathsf{Err}^\star_{\mathrm{DP}}(\{(\varepsilon_n, \delta_n)\}_{n\geq 1}) \leq \mathsf{Err}^\star_{\mathrm{lo}}(\chi + \eta).$$

*Proof.* We first outline the structure of the argument. The key point is that, under the unbiased mean-estimation constraint, the $n$-user shuffled design problem can be reduced to a *single-user* estimation problem in the same spirit as (Asi et al., 2022). After performing this reduction, the remaining task is to relate the shuffled-DP constraint to shuffle-index constraints via Lemma 2.6. This yields two single-user optimization problems, one governed by $\chi_{\mathrm{up}}(\mathcal{R})$ and the other by $\chi_{\mathrm{lo}}(\mathcal{R})$, which lead respectively to the lower and upper bounds in the statement.

Accordingly, we proceed in two steps. We begin with the reduction from the $n$-user optimization to single-user estimation, and then we prove the desired upper and lower bounds separately.

### A.2.1. REDUCTION TO SINGLE-USER ESTIMATION

We first reformulate the $n$-user design problem by replacing the shuffled-DP constraint with an explicit shuffle-index constraint on the local randomizer $\mathcal{R}$, which leads to (14) (and analogously (15)).

$$
\begin{aligned}
\mathsf{Err}^{\star,\mathrm{uni}}_{\mathrm{lo}}(\chi) := \inf_{\mathcal{R}\in\mathfrak{R}} \; &\inf_{\{\mathcal{A}_n\}_{n\geq 1}} \; \limsup_{n\to\infty} \; n \cdot \mathsf{Err}_n(\mathcal{R}, \mathcal{A}_n) \\
&\text{s.t.} \quad (\mathcal{R}^n, \mathcal{A}_n) \text{ is unbiased for all sufficiently large } n, \\
&\qquad\;\; \chi_{\mathrm{lo}}(\mathcal{R}) \geq \chi,
\end{aligned}
\tag{14}
$$

$$
\begin{aligned}
\mathsf{Err}^{\star,\mathrm{uni}}_{\mathrm{up}}(\chi) := \inf_{\mathcal{R}\in\mathfrak{R}} \; &\inf_{\{\mathcal{A}_n\}_{n\geq 1}} \; \limsup_{n\to\infty} \; n \cdot \mathsf{Err}_n(\mathcal{R}, \mathcal{A}_n) \\
&\text{s.t.} \quad (\mathcal{R}^n, \mathcal{A}_n) \text{ is unbiased for all sufficiently large } n, \\
&\qquad\;\; \chi_{\mathrm{up}}(\mathcal{R}) \geq \chi.
\end{aligned}
\tag{15}
$$

Let $\mathsf{Err}^\star_{\mathrm{lo}}(\chi)$ and $\mathsf{Err}^\star_{\mathrm{up}}(\chi)$ denote the optimal values of (5) and (6), respectively. Let $\mathsf{Err}^{\star,\mathrm{uni}}_{\mathrm{lo}}(\chi)$ and $\mathsf{Err}^{\star,\mathrm{uni}}_{\mathrm{up}}(\chi)$ denote the optimal values of (14) and (15), respectively.

We can reduce these $n$-user optimization problems to single-user estimation problems.

**Lemma A.2** (Reduction to single-user estimation). *Fix $\chi > 0$. Then*

$$\mathsf{Err}^\star_{\mathrm{lo}}(\chi) = \mathsf{Err}^{\star,\mathrm{uni}}_{\mathrm{lo}}(\chi).$$

$$\mathsf{Err}^\star_{\mathrm{up}}(\chi) = \mathsf{Err}^{\star,\mathrm{uni}}_{\mathrm{up}}(\chi).$$

The proof is given in Appendix A.2.4.

A.2.2. UPPER BOUND

Fix any $\eta > 0$. Let $(\mathcal{R}^\eta, \widehat{x}^\eta)$ be an (exact or $\zeta$-approximate) minimizer for the single-user shuffle-index constrained problem (5) at level $\chi + \eta$, i.e.,

$$\mathbb{E}_{Y \sim \mathcal{R}_x^\eta}[\widehat{x}^\eta(Y)] = x \quad \text{for all } x \in \mathbb{B}_2^d,$$
$$\chi_{\mathrm{lo}}(\mathcal{R}^\eta) \geq \chi + \eta,$$
$$\mathsf{Err}_1(\mathcal{R}^\eta, \widehat{x}^\eta) \leq \mathsf{Err}_{\mathrm{lo}}^\star(\chi + \eta) + \zeta,$$

where $\zeta > 0$ is arbitrary.

**Step 1: Privacy feasibility under** $(\varepsilon_n, \delta_n)$. Apply Lemma 2.6 (upper bound direction) to $\mathcal{R}^\eta$. Since $\varepsilon_n \to 0$, $\varepsilon_n = \omega(\sqrt{1/n})$, and $\varepsilon_n = O(\sqrt{\log n/n})$, there exists a sequence $e_n^{\mathrm{lo}}(\eta) \to 0$ such that for all sufficiently large $n$,

$$\delta_{\mathcal{S} \circ (\mathcal{R}^\eta)^n}(\varepsilon_n) \leq f_{n,\varepsilon_n}(\chi_{\mathrm{lo}}(\mathcal{R}^\eta)) \left(1 + e_n^{\mathrm{lo}}(\eta)\right). \tag{16}$$

Using $\chi_{\mathrm{lo}}(\mathcal{R}^\eta) \geq \chi + \eta$ and that $f_{n,\varepsilon}(\cdot)$ is decreasing, we obtain

$$\delta_{\mathcal{S} \circ (\mathcal{R}^\eta)^n}(\varepsilon_n) \leq f_{n,\varepsilon_n}(\chi + \eta) \left(1 + e_n^{\mathrm{lo}}(\eta)\right). \tag{17}$$

Next, compare $f_{n,\varepsilon_n}(\chi + \eta)$ and $f_{n,\varepsilon_n}(\chi)$:

$$\frac{f_{n,\varepsilon_n}(\chi + \eta)}{f_{n,\varepsilon_n}(\chi)} = \left(\frac{\chi}{\chi + \eta}\right)^3 \exp\left(-\frac{(\chi + \eta)^2 - \chi^2}{2} \varepsilon_n^2 \, n\right). \tag{18}$$

Since $\eta > 0$ is fixed and $\varepsilon_n^2 n \to \infty$ (because $\varepsilon_n = \omega(\sqrt{1/n})$), the right-hand side of (18) tends to 0.

From the definition 3.1, we have $\delta_n = f_{n,\varepsilon_n}(\chi)(1 + e_n)$ with $e_n \to 0$. Therefore, combining (17) and (18), we conclude that for all sufficiently large $n$,

$$\delta_{\mathcal{S} \circ (\mathcal{R}^\eta)^n}(\varepsilon_n) \leq \delta_n. \tag{19}$$

Hence the shuffled mechanism induced by $\mathcal{R}^\eta$ is $(\varepsilon_n, \delta_n)$-DP for all sufficiently large $n$.

**Step 2: Risk bound.** Define an $n$-user analyzer $\mathcal{A}^\eta : \mathcal{Y}^n \to \mathbb{R}^d$ by

$$\mathcal{A}^\eta(y_{1:n}) := \frac{1}{n} \sum_{i=1}^n \widehat{x}^\eta(y_i),$$

which is permutation-invariant and hence compatible with shuffling. Since $(\mathcal{R}^\eta, \widehat{x}^\eta)$ is unbiased at the single-user level, $((\mathcal{R}^\eta)^n, \mathcal{A}^\eta)$ is unbiased for the mean. Moreover, by Lemma A.2,

$$\mathsf{Err}_n(\mathcal{R}^\eta, \mathcal{A}^\eta) = \frac{1}{n} \, \mathsf{Err}_1(\mathcal{R}^\eta, \widehat{x}^\eta).$$

Combining this with (19), we see that $(\mathcal{R}^\eta, \mathcal{A}^\eta)$ is feasible for (4) (for all sufficiently large $n$), and thus

$$\mathsf{Err}_{\mathrm{DP}}^\star(\{(\varepsilon_n, \delta_n)\}_{n \geq 1}) \leq \limsup_{n \to \infty} n \mathsf{Err}_n(\mathcal{R}^\eta, \mathcal{A}^\eta) = \mathsf{Err}_1(\mathcal{R}^\eta, \widehat{x}^\eta) \leq \mathsf{Err}_{\mathrm{lo}}^\star(\chi + \eta) + \zeta.$$

Finally, letting $\zeta \downarrow 0$ proves the claim.

A.2.3. LOWER BOUND

Assume throughout that $(\varepsilon_n, \delta_n)$ is in the $\chi$-high privacy regime (Definition 3.1).

We prove the lower bound

$$\mathsf{Err}_{\mathrm{up}}^\star(\chi) \leq \mathsf{Err}_{\mathrm{DP}}^\star(\{(\varepsilon_n, \delta_n)\}_{n \geq 1}).$$

**Step 1: Any shuffled-DP feasible $\mathcal{R}$ must satisfy $\chi_{\mathrm{up}}(\mathcal{R}) \geq \chi$.** Fix any local randomizer $\mathcal{R} \in \mathfrak{R}$ and any analyzer $\mathcal{A}$ such that $(\mathcal{R}^n, \mathcal{A})$ is unbiased and the shuffled mechanism is $(\varepsilon_n, \delta_n)$-DP, i.e.,

$$\delta_{\mathcal{S} \circ \mathcal{R}^n}(\varepsilon_n) \leq \delta_n. \tag{20}$$

We claim that necessarily $\chi_{\mathrm{up}}(\mathcal{R}) \geq \chi$ for all sufficiently large $n$.

Suppose for contradiction that $\chi_{\mathrm{up}}(\mathcal{R}) < \chi$. Since $\varepsilon_n \to 0$, $\varepsilon_n = \omega(\sqrt{1/n})$, and $\varepsilon_n = O(\sqrt{\log n/n})$ by the definition of the high privacy regime, we may apply Lemma 2.6 to obtain a sequence $e_n^{\mathrm{up}} \to 0$ such that, for all sufficiently large $n$,

$$\delta_{\mathcal{S} \circ \mathcal{R}^n}(\varepsilon_n) \geq f_{n,\varepsilon_n}(\chi_{\mathrm{up}}(\mathcal{R}))\left(1 + e_n^{\mathrm{up}}\right), \tag{21}$$

where $f_{n,\varepsilon}(\chi)$ is defined in Lemma 2.6.

On the other hand, since $(\varepsilon_n, \delta_n)$ is in the $\chi$-high privacy regime, there exists a sequence $e_n \to 0$ such that, for all sufficiently large $n$,

$$\delta_n = f_{n,\varepsilon_n}(\chi)\left(1 + e_n\right). \tag{22}$$

Combining (20), (21), and (22) yields, for all sufficiently large $n$,

$$f_{n,\varepsilon_n}(\chi_{\mathrm{up}}(\mathcal{R}))\left(1 + e_n^{\mathrm{up}}\right) \leq f_{n,\varepsilon_n}(\chi)\left(1 + e_n\right). \tag{23}$$

Now consider the ratio

$$\frac{f_{n,\varepsilon_n}(\chi_{\mathrm{up}}(\mathcal{R}))}{f_{n,\varepsilon_n}(\chi)} = \left(\frac{\chi}{\chi_{\mathrm{up}}(\mathcal{R})}\right)^3 \exp\left(\frac{\chi^2 - \chi_{\mathrm{up}}(\mathcal{R})^2}{2}\varepsilon_n^2\, n\right).$$

Since $\chi_{\mathrm{up}}(\mathcal{R}) < \chi$, we have $\chi^2 - \chi_{\mathrm{up}}(\mathcal{R})^2 > 0$, and since $\varepsilon_n = \omega(\sqrt{1/n})$ we have $\varepsilon_n^2 n \to \infty$. Therefore,

$$\frac{f_{n,\varepsilon_n}(\chi_{\mathrm{up}}(\mathcal{R}))}{f_{n,\varepsilon_n}(\chi)} \longrightarrow \infty.$$

Moreover, $(1 + e_n^{\mathrm{up}})/(1 + e_n) \to 1$. Hence, for all sufficiently large $n$,

$$f_{n,\varepsilon_n}(\chi_{\mathrm{up}}(\mathcal{R}))\left(1 + e_n^{\mathrm{up}}\right) > f_{n,\varepsilon_n}(\chi)\left(1 + e_n\right),$$

which contradicts (23). This proves the claim:

$$\chi_{\mathrm{up}}(\mathcal{R}) \geq \chi \tag{24}$$

**Step 2: Reduction to the $\chi_{\mathrm{up}}$-constrained minimax risk.** By (24), every protocol feasible for (4) (i.e., unbiased and shuffled-DP at level $(\varepsilon_n, \delta_n)$) is also feasible for the shuffled-index constrained problem defining $\mathrm{Err}_{\mathrm{up}}^{\star,\mathrm{uni}}(\chi)$ (namely, the same unbiasedness constraint together with $\chi_{\mathrm{up}}(\mathcal{R}) \geq \chi$). Therefore the feasible set of (4) is contained in the feasible set of the $\chi_{\mathrm{up}}$-constrained problem, and consequently the optimal value satisfies

$$\mathrm{Err}_{\mathrm{DP}}^{\star}(\{(\varepsilon_n, \delta_n)\}_{n\geq 1}) \geq \mathrm{Err}_{\mathrm{up}}^{\star,\mathrm{uni}}(\chi). \tag{25}$$

Finally, by Lemma A.2 we have $\mathrm{Err}_{\mathrm{up}}^{\star,\mathrm{uni}}(\chi) = \mathrm{Err}_{\mathrm{up}}^{\star}(\chi)$. Substituting into (25) yields

$$\mathrm{Err}_{\mathrm{DP}}^{\star}(\{(\varepsilon_n, \delta_n)\}_{n\geq 1}) \geq \mathrm{Err}_{\mathrm{up}}^{\star}(\chi),$$

as desired.

$\square$

### A.2.4. LEMMA A.2

**Lemma A.2** (Reduction to single-user estimation)**.** *Fix $\chi > 0$. Then*

$$\mathrm{Err}_{\mathrm{lo}}^{\star}(\chi) = \mathrm{Err}_{\mathrm{lo}}^{\star,\mathrm{uni}}(\chi).$$

$$\mathrm{Err}_{\mathrm{up}}^{\star}(\chi) = \mathrm{Err}_{\mathrm{up}}^{\star,\mathrm{uni}}(\chi).$$

*Proof.* The reduction is in the same spirit as the canonicalization results of Asi et al. (Asi et al., 2022) for unbiased local protocols. We first establish two auxiliary results: Lemma A.3 and Lemma A.4, and then use them to prove Lemma A.2.

**Lemma A.3** (Post-processing for shuffle indices). *Let $\mathcal{R} \in \mathfrak{R}$ be a local randomizer that admits a blanket distribution $\mathcal{R}_{\mathrm{BG}}$ with blanket mass $\gamma \in (0, 1]$, i.e., for every $x \in \mathcal{X}_\perp$ and a.e. $y \in \mathcal{Y}$,*

$$\mathcal{R}_x(y) = \gamma \, \mathcal{R}_{\mathrm{BG}}(y) + (1 - \gamma) \, Q_x(y)$$

*for some family of densities $\{Q_x\}_{x \in \mathcal{X}_\perp}$. Let $\mathcal{K} : \mathcal{Y} \to \mathcal{Z}$ be an arbitrary (possibly randomized) post-processing map, i.e., a Markov kernel from $\mathcal{Y}$ to $\mathcal{Z}$. Define the post-processed randomizer*

$$\mathcal{R}' := \mathcal{K} \circ \mathcal{R} : \mathcal{X}_\perp \to \mathcal{Z}, \qquad \mathcal{R}'_x := \mathcal{K} \circ \mathcal{R}_x.$$

*Then $\mathcal{R}'$ admits a blanket distribution $\mathcal{R}'_{\mathrm{BG}} := \mathcal{K} \circ \mathcal{R}_{\mathrm{BG}}$ with blanket mass at least $\gamma$, and moreover its shuffle indices satisfy*

$$\chi_{\mathrm{up}}(\mathcal{R}') \geq \chi_{\mathrm{up}}(\mathcal{R}), \qquad \chi_{\mathrm{lo}}(\mathcal{R}') \geq \chi_{\mathrm{lo}}(\mathcal{R}).$$

*Proof.* We first show that $\mathcal{R}'$ admits a blanket of mass $\gamma$. Applying $\mathcal{K}$ to the blanket decomposition of $\mathcal{R}_x$ yields

$$\mathcal{R}'_x = \mathcal{K} \circ \mathcal{R}_x = \gamma \, (\mathcal{K} \circ \mathcal{R}_{\mathrm{BG}}) + (1 - \gamma) \, (\mathcal{K} \circ Q_x).$$

Thus, $\mathcal{R}'_{\mathrm{BG}} := \mathcal{K} \circ \mathcal{R}_{\mathrm{BG}}$ is a valid blanket distribution of $\mathcal{R}'$ with blanket mass $\gamma$ (and hence the maximal blanket mass $\gamma'$ of $\mathcal{R}'$ satisfies $\gamma' \geq \gamma$).

We next prove the monotonicity of the lower shuffle index. Fix $x \simeq x'$ and consider the generalized privacy amplification random variable with reference $\mathcal{R}_{\mathrm{BG}}$:

$$\ell_0(Y; x, x', \mathcal{R}_{\mathrm{BG}}) := \frac{\mathcal{R}_x(Y) - \mathcal{R}_{x'}(Y)}{\mathcal{R}_{\mathrm{BG}}(Y)}, \qquad Y \sim \mathcal{R}_{\mathrm{BG}}.$$

Let $Z \sim \mathcal{R}'_{\mathrm{BG}}$ be obtained by sampling $Y \sim \mathcal{R}_{\mathrm{BG}}$ and then applying $Z \mid Y \sim \mathcal{K}(\cdot \mid Y)$. Define analogously

$$\ell'_0(Z; x, x', \mathcal{R}'_{\mathrm{BG}}) := \frac{\mathcal{R}'_x(Z) - \mathcal{R}'_{x'}(Z)}{\mathcal{R}'_{\mathrm{BG}}(Z)}.$$

Then, for $\mathcal{R}_{\mathrm{BG}}$-a.e. $y$ and $\mathcal{R}'_{\mathrm{BG}}$-a.e. $z$,

$$\ell'_0(z; x, x', \mathcal{R}'_{\mathrm{BG}}) = \mathbb{E}[\ell_0(Y; x, x', \mathcal{R}_{\mathrm{BG}}) \mid Z = z].$$

Indeed, writing $\mathcal{K}(z \mid y)$ for the kernel density, we have

$$\mathcal{R}'_x(z) - \mathcal{R}'_{x'}(z) = \int \mathcal{K}(z \mid y) \big( \mathcal{R}_x(y) - \mathcal{R}_{x'}(y) \big) \, dy, \qquad \mathcal{R}'_{\mathrm{BG}}(z) = \int \mathcal{K}(z \mid y) \mathcal{R}_{\mathrm{BG}}(y) \, dy,$$

and the claim follows by Bayes' rule.

Therefore, by the law of total variance,

$$\mathrm{Var}(\ell'_0(Z; x, x', \mathcal{R}'_{\mathrm{BG}})) = \mathrm{Var}(\mathbb{E}[\ell_0(Y; x, x', \mathcal{R}_{\mathrm{BG}}) \mid Z]) \leq \mathrm{Var}(\ell_0(Y; x, x', \mathcal{R}_{\mathrm{BG}})).$$

Combining this with $\gamma' \geq \gamma$ yields

$$\frac{1}{\gamma'} \mathrm{Var}(\ell'_0(Z; x, x', \mathcal{R}'_{\mathrm{BG}})) \leq \frac{1}{\gamma} \mathrm{Var}(\ell_0(Y; x, x', \mathcal{R}_{\mathrm{BG}})).$$

Taking the supremum over $x \simeq x'$ and the square root gives $\chi_{\mathrm{lo}}(\mathcal{R}') \geq \chi_{\mathrm{lo}}(\mathcal{R})$.

The proof for the upper shuffle index is analogous. For any fixed $x \in \mathcal{X}$, let $Y \sim \mathcal{R}_x$ and obtain $Z$ by applying the same post-processing kernel $\mathcal{K}$. Then, defining

$$\ell_{0,x}(Y; x_1, x'_1) := \frac{\mathcal{R}_{x_1}(Y) - \mathcal{R}_{x'_1}(Y)}{\mathcal{R}_x(Y)}, \qquad \ell'_{0,x}(Z; x_1, x'_1) := \frac{\mathcal{R}'_{x_1}(Z) - \mathcal{R}'_{x'_1}(Z)}{\mathcal{R}'_x(Z)},$$

one similarly has

$$\ell'_{0,x}(Z; x_1, x'_1) = \mathbb{E}[\ell_{0,x}(Y; x_1, x'_1) \mid Z],$$

and hence

$$\mathrm{Var}\big(\ell'_{0,x}(Z; x_1, x'_1)\big) \le \mathrm{Var}(\ell_{0,x}(Y; x_1, x'_1)).$$

Taking the supremum over $x_1 \simeq x'_1$ and $x \in \mathcal{X}$, and then the square root, concludes that $\chi_{\mathrm{up}}(\mathcal{R}') \ge \chi_{\mathrm{up}}(\mathcal{R})$. $\qquad \square$

**Lemma A.4** (Additivization via a Markov kernel). *Fix $n \ge 2$. Let $\mathcal{R}$ be any local randomizer and let $\mathcal{A}$ be any unbiased estimator in the single-message shuffle model. Then there exists a Markov kernel $K : \mathcal{Y} \to \mathbb{R}^d$ such that, letting $\mathcal{A}^+(z_{1:n}) := \frac{1}{n} \sum_{i=1}^n z_i$, the estimator $\mathcal{A}^+$ is unbiased with respect to the local randomizer $K \circ \mathcal{R} : \mathcal{X} \to \mathbb{R}^d$, and moreover,*

$$\mathsf{Err}_n(K \circ \mathcal{R}, \mathcal{A}^+) \le \mathsf{Err}_n(\mathcal{R}, \mathcal{A}).$$

*Proof.* Define the composed analyzer $\widetilde{\mathcal{A}} := \mathcal{A} \circ \mathcal{S}$. Since shuffling is a post-processing operation, the single-message shuffled protocol $(\mathcal{R}, \mathcal{A})$ can be equivalently viewed as the (non-shuffled) local protocol $(\mathcal{R}, \widetilde{\mathcal{A}})$ with the same output distribution. In particular, unbiasedness is preserved: for all datasets $x_{1:n} \in \mathcal{X}^n$,

$$\mathbb{E}\Big[\widetilde{\mathcal{A}}(\mathcal{R}^n(x_{1:n}))\Big] = \mathbb{E}[\mathcal{A}(\mathcal{S}(\mathcal{R}^n(x_{1:n})))] = \frac{1}{n} \sum_{i=1}^n x_i.$$

Therefore, we may invoke a canonicalization (additivization) argument in the spirit of Proposition 3.3 of Asi et al. (2022). Concretely, for any unbiased analyzer $\widetilde{\mathcal{A}}$, there exists a Markov kernel $K : \mathcal{Y} \to \mathbb{R}^d$ such that, letting $\mathcal{R}' := K \circ \mathcal{R}$ and $\mathcal{A}^+(z_{1:n}) := \frac{1}{n} \sum_{i=1}^n z_i$, the estimator $\mathcal{A}^+$ is unbiased with respect to $\mathcal{R}'$ and satisfies

$$\mathsf{Err}_n(\mathcal{R}', \mathcal{A}^+) \le \mathsf{Err}_n(\mathcal{R}, \widetilde{\mathcal{A}}).$$

Unlike (Asi et al., 2022), we do not require $\mathcal{R}$ to satisfy $\varepsilon$-LDP; the canonicalization construction we use relies only on unbiasedness. The only closure property needed under post-processing concerns the shuffle-index constraints, which is guaranteed by Lemma A.3. $\qquad \square$

**Proof of Lemma A.2** We prove the claim for the $\chi_{\mathrm{lo}}$-constrained problem; the proof for $\chi_{\mathrm{up}}$ is identical by using the corresponding monotonicity in Lemma A.3.

Recall that $\mathsf{Err}^\star_{n,\mathrm{lo}}(\chi)$ is the optimal value of (14) and $\mathsf{Err}^\star_{\mathrm{lo}}(\chi)$ is the optimal value of (5).

**Step 1: $\le$ direction.** Let $(\mathcal{R}, \widehat{x})$ be any feasible solution to the single-user problem (5), i.e., $\mathbb{E}_{Y \sim \mathcal{R}_x}[\widehat{x}(Y)] = x$ for all $x \in \mathbb{B}_2^d$ and $\chi_{\mathrm{lo}}(\mathcal{R}) \ge \chi$. Define an $n$-user protocol by applying $\mathcal{R}$ independently to each user and using the additive estimator

$$\mathcal{A}(y_{1:n}) := \frac{1}{n} \sum_{i=1}^n \widehat{x}(y_i), \qquad y_{1:n} \in \mathcal{Y}^n,$$

(which is permutation-invariant and hence unaffected by shuffling). Then the resulting estimator is unbiased:

$$\mathbb{E}[\mathcal{A}(\mathcal{S}(\mathcal{R}^n(x_{1:n})))] = \frac{1}{n} \sum_{i=1}^n \mathbb{E}[\widehat{x}(Y_i)] = \frac{1}{n} \sum_{i=1}^n x_i.$$

Moreover, writing $Y_i \sim \mathcal{R}_{x_i}$ independently and setting $\Delta_i := \widehat{x}(Y_i) - x_i$, we have $\mathbb{E}[\Delta_i] = 0$ and hence

$$\mathbb{E}\Big\|\mathcal{A}(Y_{1:n}) - \frac{1}{n} \sum_{i=1}^n x_i\Big\|_2^2 = \mathbb{E}\Big\|\frac{1}{n} \sum_{i=1}^n \Delta_i\Big\|_2^2 = \frac{1}{n^2} \sum_{i=1}^n \mathbb{E}\|\Delta_i\|_2^2,$$

where the cross terms vanish by independence and mean-zero. Taking the supremum over $x_{1:n} \in (\mathbb{B}_2^d)^n$ yields

$$\mathsf{Err}_n(\mathcal{R}, \mathcal{A}) = \frac{1}{n} \sup_{x \in \mathbb{B}_2^d} \mathbb{E}_{Y \sim \mathcal{R}_x}\big[\|\widehat{x}(Y) - x\|_2^2\big] = \frac{1}{n} \mathsf{Err}_1(\mathcal{R}, \widehat{x}).$$

Since $(\mathcal{R}, \mathcal{A})$ is feasible for (14), we obtain

$$\mathsf{Err}^\star_{n,\mathrm{lo}}(\chi) \le \mathsf{Err}^\star_{\mathrm{lo}}(\chi).$$

**Step 2: $\geq$ direction.** Let $(\mathcal{R}, \mathcal{A})$ be any feasible $n$-user solution to (14), i.e., $\mathcal{A}$ is unbiased for $\mathcal{S} \circ \mathcal{R}^n$ and $\chi_{\mathrm{lo}}(\mathcal{R}) \geq \chi$. Define the non-shuffled analyzer $\widetilde{\mathcal{A}} := \mathcal{A} \circ \mathcal{S}$. Since shuffling is post-processing, the pair $(\mathcal{R}, \widetilde{\mathcal{A}})$ induces the same output distribution and remains unbiased:

$$\mathbb{E}\left[\widetilde{\mathcal{A}}(\mathcal{R}^n(x_{1:n}))\right] = \mathbb{E}[\mathcal{A}(\mathcal{S}(\mathcal{R}^n(x_{1:n})))] = \frac{1}{n}\sum_{i=1}^{n} x_i.$$

Applying Lemma A.4 to $(\mathcal{R}, \widetilde{\mathcal{A}})$, we obtain a Markov kernel $K : \mathcal{Y} \to \mathbb{R}^d$ such that, letting $\mathcal{R}' := K \circ \mathcal{R}$ and $\mathcal{A}^+(z_{1:n}) := \frac{1}{n}\sum_{i=1}^{n} z_i$, the estimator $\mathcal{A}^+$ is unbiased with respect to $\mathcal{R}'$ and

$$\mathsf{Err}_n(\mathcal{R}', \mathcal{A}^+) \leq \mathsf{Err}_n(\mathcal{R}, \mathcal{A}). \tag{26}$$

By Lemma A.3,

$$\chi_{\mathrm{lo}}(\mathcal{R}') = \chi_{\mathrm{lo}}(K \circ \mathcal{R}) \geq \chi_{\mathrm{lo}}(\mathcal{R}) \geq \chi,$$

so $(\mathcal{R}', \mathcal{A}^+)$ is feasible for (14).

Now let $Z_i \sim \mathcal{R}'_{x_i}$ be independent and define $\Xi_i := Z_i - x_i$. Unbiasedness implies $\mathbb{E}[\Xi_i] = 0$, and thus

$$\mathbb{E}\left\|\mathcal{A}^+(Z_{1:n}) - \frac{1}{n}\sum_{i=1}^{n} x_i\right\|_2^2 = \mathbb{E}\left\|\frac{1}{n}\sum_{i=1}^{n}\Xi_i\right\|_2^2 = \frac{1}{n^2}\sum_{i=1}^{n}\mathbb{E}\|\Xi_i\|_2^2,$$

and hence

$$\mathsf{Err}_n(\mathcal{R}', \mathcal{A}^+) = \frac{1}{n}\sup_{x \in \mathbb{B}_2^d}\mathbb{E}_{Z \sim \mathcal{R}'_x}\left[\|Z - x\|_2^2\right] = \frac{1}{n}\mathsf{Err}_1(\mathcal{R}', \mathrm{id}),$$

where $\mathrm{id} : \mathbb{R}^d \to \mathbb{R}^d$ denotes the identity estimator. Since $\mathrm{id}$ is an admissible unbiased estimator for $\mathcal{R}'$, the single-user optimum satisfies

$$\mathsf{Err}_{\mathrm{lo}}^{\star}(\chi) \leq \mathsf{Err}_1(\mathcal{R}', \mathrm{id}).$$

Combining the last two displays with (26) gives

$$\mathsf{Err}_n(\mathcal{R}, \mathcal{A}) \geq \mathsf{Err}_n(\mathcal{R}', \mathcal{A}^+) = \frac{1}{n}\mathsf{Err}_1(\mathcal{R}', \mathrm{id}) \geq \frac{1}{n}\mathsf{Err}_{\mathrm{lo}}^{\star}(\chi).$$

Taking the infimum over all feasible $(\mathcal{R}, \mathcal{A})$ for (14) yields

$$\mathsf{Err}_{n,\mathrm{lo}}^{\star}(\chi) \geq \mathsf{Err}_{\mathrm{lo}}^{\star}(\chi).$$

**Conclusion.** Combining the $\leq$ and $\geq$ directions proves

$$\mathsf{Err}_{n,\mathrm{lo}}^{\star}(\chi) = \mathsf{Err}_{\mathrm{lo}}^{\star}(\chi).$$

The statement for $\mathsf{Err}_{n,\mathrm{up}}^{\star}(\chi)$ follows by the same argument, replacing $\chi_{\mathrm{lo}}$ by $\chi_{\mathrm{up}}$ throughout.

$\square$

## A.3. Theorem 3.5

**Theorem 3.5.** *Let $\mathcal{R} \in \mathfrak{R}$ be a local randomizer with $\chi_{\mathrm{up}}(\mathcal{R}) < \infty$ and $\widehat{x}$ be any unbiased estimator. Then,*

$$\mathsf{Err}_1(\mathcal{R}, \widehat{x}) \geq d\,\chi_{\mathrm{up}}(\mathcal{R})^2.$$

*Proof.* Let $\mathcal{R} : \mathcal{X}_{\perp} \to \mathcal{Y}$ be a local randomizer, and let $\widehat{x} : \mathcal{Y} \to \mathbb{R}^d$ be an unbiased estimator in the sense that

$$\mathbb{E}_{Y \sim \mathcal{R}_x}\left[\widehat{x}(Y)\right] = x \qquad \text{for all } x \in \mathcal{X}_{\perp},$$

where we interpret $\perp$ as contributing 0 to the mean and thus require $\mathbb{E}_{Y \sim \mathcal{R}_{\perp}}[\widehat{x}(Y)] = 0$.

Fix an arbitrary reference point $x \in \mathcal{X}$. By mutual absolute continuity due to $\mathcal{R} \in \mathfrak{R}$, for every $x' \in \mathcal{X}_\perp$ the Radon–Nikodym derivative

$$w_{x';x}(y) := \frac{d\mathcal{R}_{x'}}{d\mathcal{R}_x}(y)$$

is well-defined $\mathcal{R}_x$-a.s. For $x_1, x_1' \in \mathcal{X}_\perp$, define

$$\ell_{x_1,x_1';x}(y) := w_{x_1;x}(y) - w_{x_1';x}(y).$$

Then $\mathbb{E}_{Y \sim \mathcal{R}_x}[\ell_{x_1,x_1';x}(Y)] = 0$. Moreover, by a change of measure and unbiasedness, for any $x' \in \mathcal{X}_\perp$,

$$\mathbb{E}_{Y \sim \mathcal{R}_{x'}}[\widehat{x}(Y)] = \mathbb{E}_{Y \sim \mathcal{R}_x}\big[\widehat{x}(Y)\, w_{x';x}(Y)\big] = x'.$$

Subtracting the identities for $x'$ equal to $x_1$ and $x_1'$ yields

$$\mathbb{E}_{Y \sim \mathcal{R}_x}\big[\widehat{x}(Y)\, \ell_{x_1,x_1';x}(Y)\big] = x_1 - x_1'. \tag{27}$$

Let $\varphi \in (\mathbb{R}^d)^*$ be any linear functional. Applying $\varphi$ to (27) gives

$$\mathbb{E}_{Y \sim \mathcal{R}_x}\Big[\varphi(\widehat{x}(Y))\, \ell_{x_1,x_1';x}(Y)\Big] = \varphi(x_1 - x_1').$$

Since $\mathbb{E}_{\mathcal{R}_x}[\ell_{x_1,x_1';x}(Y)] = 0$, we may center $\varphi(\widehat{x}(Y))$ to obtain

$$\mathbb{E}_{Y \sim \mathcal{R}_x}\Big[\big(\varphi(\widehat{x}(Y)) - \mathbb{E}_{\mathcal{R}_x}[\varphi(\widehat{x}(Y))]\big)\, \ell_{x_1,x_1';x}(Y)\Big] = \varphi(x_1 - x_1').$$

By Cauchy–Schwarz,

$$\big|\varphi(x_1 - x_1')\big| \le \Big(\mathrm{Var}_{Y \sim \mathcal{R}_x}\big(\varphi(\widehat{x}(Y))\big)\Big)^{1/2} \Big(\mathrm{Var}_{Y \sim \mathcal{R}_x}\big(\ell_{x_1,x_1';x}(Y)\big)\Big)^{1/2},$$

and hence

$$\mathrm{Var}_{Y \sim \mathcal{R}_x}\big(\varphi(\widehat{x}(Y))\big) \ge \frac{\big(\varphi(x_1 - x_1')\big)^2}{\mathrm{Var}_{Y \sim \mathcal{R}_x}\big(\ell_{x_1,x_1';x}(Y)\big)}. \tag{28}$$

We now identify $\ell_{x_1,x_1';x}$ with the generalized privacy amplification random variable at $\varepsilon = 0$:

$$\ell_{x_1,x_1';x}(y) = \frac{\mathcal{R}_{x_1}(y) - \mathcal{R}_{x_1'}(y)}{\mathcal{R}_x(y)} = \ell_0(y; x_1, x_1', \mathcal{R}_x).$$

Thus, by the definition of the upper shuffle index,

$$\sup_{x_1 \simeq x_1' \in \mathcal{X}_\perp}\ \sup_{x \in \mathcal{X}} \mathrm{Var}_{Y \sim \mathcal{R}_x}[\ell_0(Y; x_1, x_1', \mathcal{R}_x)] \le \frac{1}{\chi_{\mathrm{up}}(\mathcal{R})^2},$$

or equivalently, for every $x \in \mathcal{X}$ and every neighboring pair $x_1 \simeq x_1'$,

$$\mathrm{Var}_{Y \sim \mathcal{R}_x}[\ell_0(Y; x_1, x_1', \mathcal{R}_x)] \le \frac{1}{\chi_{\mathrm{up}}(\mathcal{R})^2}. \tag{29}$$

Next, fix $j \in [d]$ and take $\varphi(v) = v_j$ in (28). Choose a zero-out neighboring pair $(x_1, x_1') = (e_j, \perp)$, where $e_j$ is the $j$-th standard basis vector (note that $e_j \in \mathbb{B}_2^d$ and $e_j \simeq \perp$). Then $\varphi(x_1 - x_1') = 1$, and combining (28) with (29) yields

$$\mathrm{Var}_{Y \sim \mathcal{R}_x}\big(\widehat{x}_j(Y)\big) \ge \chi_{\mathrm{up}}(\mathcal{R})^2 \qquad \text{for all } x \in \mathcal{X}. \tag{30}$$

Finally, since $\widehat{x}$ is unbiased, for any $x \in \mathcal{X}$ we have

$$\mathbb{E}_{Y \sim \mathcal{R}_x}\big[\|\widehat{x}(Y) - x\|_2^2\big] = \sum_{j=1}^d \mathbb{E}_{Y \sim \mathcal{R}_x}\big[(\widehat{x}_j(Y) - x_j)^2\big] = \sum_{j=1}^d \mathrm{Var}_{Y \sim \mathcal{R}_x}\big(\widehat{x}_j(Y)\big),$$

and therefore by (30),

$$\mathbb{E}_{Y \sim \mathcal{R}_x}\left[\|\widehat{x}(Y) - x\|_2^2\right] \geq \sum_{j=1}^{d} \chi_{\mathrm{up}}(\mathcal{R})^2 = d\,\chi_{\mathrm{up}}(\mathcal{R})^2.$$

Taking the supremum over $x \in \mathbb{B}_2^d \subseteq \mathcal{X}$ gives

$$\mathrm{Err}_1(\mathcal{R}, \widehat{x}) = \sup_{x \in \mathbb{B}_2^d} \mathbb{E}_{Y \sim \mathcal{R}_x}\left[\|\widehat{x}(Y) - x\|_2^2\right] \geq d\,\chi_{\mathrm{up}}(\mathcal{R})^2,$$

which proves the claim. $\qquad\square$

### A.4. Proposition 3.7

**Proposition 3.7.** *Consider the input domain $\mathcal{X} = \mathbb{S}^{d-1} \subset \mathbb{R}^d$ for PrivUnit where $d > 1$. Let $\mathrm{PrivUnit}(p, \theta)$ be the PrivUnit local randomizer, and let $\chi_{\mathrm{lo}} := \chi_{\mathrm{lo}}(\mathrm{PrivUnit}(p, \theta))$. For any choice of $\{\theta_d\}_{d \geq 2}$, let $\widehat{x}_d$ be an unbiased estimator for $\mathrm{PrivUnit}(p, \theta_d)$. Then, $\chi_{\mathrm{lo}} \to \infty$,*

$$\mathrm{Err}_1(\mathrm{PrivUnit}(p, \theta_d), \widehat{x}_d) = C(\theta_d, d)d\chi_{\mathrm{lo}}^2\left(1 + O(\chi_{\mathrm{lo}}^{-1})\right)$$

*for some quantity $C(\theta_d, d) > 0$ depending only on $(\theta_d, d)$. Moreover, the best achievable leading constant is bounded away from $1$:*

$$\liminf_{d \to \infty} \inf_{\theta \in [-1,1]} C(\theta, d) \geq \frac{\pi}{2}.$$

*Proof.*

**Definition A.5** (PrivUnit local randomizer (Duchi et al., 2013; Bhowmick et al., 2019))**.** Fix $d \geq 2$ and let the input domain be $\mathcal{X} = \mathbb{S}^{d-1} \subset \mathbb{R}^d$. Fix parameters $p \in [0, 1]$ and $\theta \in [-1, 1]$. For an input $v \in \mathbb{S}^{d-1}$ define the spherical cap

$$C_\theta(v) := \{u \in \mathbb{S}^{d-1} : \langle u, v \rangle \geq \theta\}.$$

Let $U \sim \mathrm{Unif}(\mathbb{S}^{d-1})$ and define

$$q(\theta, d) := \Pr\left(\langle U, v \rangle < \theta\right), \qquad 1 - q(\theta, d) = \Pr\left(U \in C_\theta(v)\right),$$

which depends only on $(\theta, d)$ by rotational symmetry. The *PrivUnit* local randomizer $\mathrm{PrivUnit}(p, \theta)$ is the map $\mathcal{R} : \mathbb{S}^{d-1} \to \mathbb{S}^{d-1}$ defined by: given input $v \in \mathbb{S}^{d-1}$, output a random $Y \in \mathbb{S}^{d-1}$ such that

$$Y \sim \begin{cases} \mathrm{Unif}(C_\theta(v)), & \text{with probability } p, \\ \mathrm{Unif}(C_\theta(v)^c), & \text{with probability } 1 - p, \end{cases}$$

where $\mathrm{Unif}(A)$ denotes the uniform distribution on a measurable set $A \subseteq \mathbb{S}^{d-1}$ with respect to the uniform probability measure on $\mathbb{S}^{d-1}$.

Fix $d \geq 2$ and a threshold $\theta \in [-1, 1]$. Let $v \in \mathbb{S}^{d-1}$ be an arbitrary input and let $U \sim \mathrm{Unif}(\mathbb{S}^{d-1})$. Write

$$T := \langle U, v \rangle, \qquad q := \Pr(T < \theta), \qquad 1 - q = \Pr(T \geq \theta).$$

Define the (cap) conditional first moment

$$\alpha := \mathbb{E}[T \mid T \geq \theta].$$

Similarly, define

$$\beta = \mathbb{E}[T \mid T < \theta].$$

By rotational symmetry, $\mathbb{E}[U \mid T \geq \theta]$ and $\mathbb{E}[U \mid T < \theta]$ lie in $\mathrm{span}(v)$, so

$$\mathbb{E}[U \mid T \geq \theta] = \alpha\, v, \qquad \mathbb{E}[U \mid T < \theta] = \beta\, v.$$

Since $\mathbb{E}[T] = 0$, we have $(1 - q)\alpha + q\beta = 0$, hence

$$\beta = -\frac{1 - q}{q}\alpha. \tag{31}$$

**Step 1: PrivUnit and its mean.** Fix $p \in [0, 1]$. PrivUnit$(p, \theta)$ outputs $Y$ distributed as $U \mid (T \geq \theta)$ with probability $p$ and as $U \mid (T < \theta)$ with probability $1 - p$. Therefore,

$$\mathbb{E}[Y] = p\,\alpha v + (1 - p)\,\beta v = \left(p\alpha - (1 - p)\frac{1 - q}{q}\alpha\right)v = m\,v,$$

where, defining $\Delta := p + q - 1$,

$$m = \alpha\,\frac{\Delta}{q}. \tag{32}$$

**Step 2: Reduction to linear unbiased estimators.** Let $\widehat{x} : \mathbb{S}^{d-1} \to \mathbb{R}^d$ be any (possibly randomized) estimator such that $\mathbb{E}[\widehat{x}(Y) \mid v] = v$ for all $v \in \mathbb{S}^{d-1}$. We show that, without loss of generality, we may restrict attention to estimators of the form $\widehat{x}(y) = a\,y$ for a scalar $a$.

*(i) Orthogonal equivariance of PrivUnit.* For any orthogonal matrix $U \in O(d)$, PrivUnit$(p, \theta)$ satisfies the equivariance relation

$$\mathcal{R}_{Uv} \overset{d}{=} U\,\mathcal{R}_v,$$

i.e., if $Y \sim \mathcal{R}_v$ then $UY \sim \mathcal{R}_{Uv}$. (This holds because the spherical cap condition $\langle u, v \rangle \geq \theta$ is preserved under orthogonal transformations.)

*(ii) Symmetrization of the estimator.* Let $U \sim \mathrm{Haar}(O(d))$ be independent of everything else, and define the symmetrized estimator

$$\widetilde{x}(y) := \mathbb{E}_U\big[U^\top \widehat{x}(Uy)\big].$$

This definition depends only on the observation $y$ and does not depend on the unknown input $v$.

*Unbiasedness is preserved.* For any $v \in \mathbb{S}^{d-1}$ and $Y \sim \mathcal{R}_v$, let $Y' \sim \mathcal{R}_{Uv}$. By (i), we may couple so that $Y' = UY$. Then

$$\begin{aligned}
\mathbb{E}[\widetilde{x}(Y) \mid v] &= \mathbb{E}_U\big[U^\top \mathbb{E}[\widehat{x}(UY) \mid v, U]\big] \\
&= \mathbb{E}_U\big[U^\top \mathbb{E}[\widehat{x}(Y') \mid Uv]\big] = \mathbb{E}_U[U^\top(Uv)] = v,
\end{aligned}$$

so $\widetilde{x}$ is unbiased whenever $\widehat{x}$ is unbiased.

*Risk does not increase.* By Jensen's inequality and orthogonality of $U$, for any $v \in \mathbb{S}^{d-1}$,

$$\begin{aligned}
\mathbb{E}\big[\|\widetilde{x}(Y) - v\|_2^2 \mid v\big] &= \mathbb{E}\Big[\big\|\mathbb{E}_U[U^\top \widehat{x}(UY) - v]\big\|_2^2 \,\Big|\, v\Big] \\
&\leq \mathbb{E}_U\,\mathbb{E}\big[\|U^\top \widehat{x}(UY) - v\|_2^2 \mid v, U\big] \\
&= \mathbb{E}_U\,\mathbb{E}\big[\|\widehat{x}(UY) - Uv\|_2^2 \mid v, U\big].
\end{aligned}$$

Using $UY \sim \mathcal{R}_{Uv}$ from (i) and the fact that the map $w \mapsto \mathbb{E}[\|\widehat{x}(Y) - w\|_2^2 \mid w]$ is constant over $w \in \mathbb{S}^{d-1}$ by the same symmetry, we conclude that

$$\sup_{v \in \mathbb{S}^{d-1}} \mathbb{E}\big[\|\widetilde{x}(Y) - v\|_2^2 \mid v\big] \;\leq\; \sup_{v \in \mathbb{S}^{d-1}} \mathbb{E}\big[\|\widehat{x}(Y) - v\|_2^2 \mid v\big].$$

Hence, for risk, it is without loss of generality to assume that the estimator is *orthogonally equivariant*, i.e.,

$$\widetilde{x}(Uy) = U\,\widetilde{x}(y) \qquad \forall\,U \in O(d),\; \forall\,y \in \mathbb{S}^{d-1}.$$

*(iii) Form of an orthogonally equivariant map on the sphere.* Let $e_1$ be the first basis vector and let $H := \{U \in O(d) : Ue_1 = e_1\}$ be its stabilizer subgroup. Equivariance implies $\widetilde{x}(e_1) = \widetilde{x}(Ue_1) = U\widetilde{x}(e_1)$ for all $U \in H$. The only vectors fixed by all $U \in H$ are multiples of $e_1$, so $\widetilde{x}(e_1) = ae_1$ for some scalar $a$. For a general $y \in \mathbb{S}^{d-1}$, pick $V \in O(d)$ with $Ve_1 = y$; then

$$\widetilde{x}(y) = \widetilde{x}(Ve_1) = V\widetilde{x}(e_1) = a\,Ve_1 = a\,y.$$

Therefore, without loss of generality, we may assume $\widehat{x}(y) = a\,y$.

Finally, imposing unbiasedness gives $a\,\mathbb{E}[Y \mid v] = v$, and since $\mathbb{E}[Y \mid v] = mv$ (from Step 1), we obtain $a = 1/m$. Hence it suffices to analyze the estimator $\widehat{x}(Y) = Y/m$.

**Step 3: Exact $\ell_2$ risk under the optimal unbiased estimator.** Since $\|Y\|_2 = 1$ almost surely and $\|v\|_2 = 1$, we compute

$$\mathsf{Err}_1(\mathrm{PrivUnit}(p,\theta), \widehat{x}) = \mathbb{E}\big[\|Y/m - v\|_2^2\big]$$

$$= \frac{1}{m^2}\,\mathbb{E}\|Y\|_2^2 - \frac{2}{m}\,\langle \mathbb{E}[Y], v\rangle + \|v\|_2^2$$

$$= \frac{1}{m^2} - 2 + 1 = \frac{1}{m^2} - 1.$$

In the high privacy regime, we have $m \to 0$, hence

$$\mathsf{Err}_1(\mathrm{PrivUnit}(p,\theta), \widehat{x}) = \frac{1}{m^2}\big(1 + o(1)\big). \tag{33}$$

Combining (32) and (33) yields

$$\mathsf{Err}_1(\mathrm{PrivUnit}(p,\theta), \widehat{x}) = \frac{q^2}{\alpha^2\,\Delta^2}\big(1 + o(1)\big). \tag{34}$$

**Step 4: Relating $\Delta$ to the lower shuffle index $\chi_{\mathrm{lo}}$.** Let $\mu$ denote the uniform probability measure on $\mathbb{S}^{d-1}$. With respect to $\mu$, the output density of $\mathrm{PrivUnit}(p,\theta)$ at input $v$ takes two values:

$$f_v(u) = \begin{cases} h := \dfrac{p}{1-q}, & u \in C_\theta(v), \\[2mm] \ell := \dfrac{1-p}{q}, & u \in C_\theta(v)^c. \end{cases}$$

Recall we adopt zero-out adjacency and set $\mathcal{R}_\perp := \mathcal{R}_{\mathrm{BG}} := \mu$. Then for neighboring inputs $(v, \perp)$ we have $\ell_0(u; v, \perp, \mathcal{R}_{\mathrm{BG}}) = f_v(u) - 1$. A direct calculation using $\mathbb{E}_\mu[f_v(U)] = 1$ shows

$$\mathrm{Var}_{U\sim\mu}[f_v(U) - 1] = (1-q)\Big(\frac{p - (1-q)}{1-q}\Big)^2 + q\Big(\frac{(1-p) - q}{q}\Big)^2 = \frac{\Delta^2}{q(1-q)}.$$

Let $\gamma$ be the blanket mass of PrivUnit with blanket distribution $\mu$. By definition of $\chi_{\mathrm{lo}}$,

$$\chi_{\mathrm{lo}}^2 = \frac{1}{\frac{1}{\gamma}\mathrm{Var}_{U\sim\mu}[f_v(U) - 1]} = \frac{\gamma\,q(1-q)}{\Delta^2}, \qquad \text{equivalently} \qquad \Delta^2 = \frac{\gamma\,q(1-q)}{\chi_{\mathrm{lo}}^2}. \tag{35}$$

Substituting (35) into (34) gives

$$\mathsf{Err}_1(\mathrm{PrivUnit}(p,\theta), \widehat{x}) = \frac{q}{1-q} \cdot \frac{1}{\gamma\,\alpha^2}\,\chi_{\mathrm{lo}}^2\big(1 + O(\chi_{\mathrm{lo}}^{-1})\big). \tag{36}$$

**Step 5: High privacy scaling and the leading constant.** The high privacy regime for PrivUnit corresponds to $\Delta \to 0$, i.e., $p \to 1 - q$, which makes the output distribution approach $\mu$. In this regime $h \to 1$ and $\ell \to 1$, hence the maximal blanket mass satisfies $\gamma \to 1$. Therefore the leading constant in (36) is

$$C(\theta, d) := \frac{q}{1-q} \cdot \frac{1}{d\,\alpha^2}, \qquad \text{so that} \qquad \mathsf{Err}_1(\mathrm{PrivUnit}(p,\theta), \widehat{x}) = C(\theta, d)\, d\, \chi_{\mathrm{lo}}^2\big(1 + o(1)\big).$$

This proves the first displayed claim of the proposition.

**Step 6: Asymptotic lower bound on the best constant over $\theta$ as $d \to \infty$.** To optimize over $\theta$ in high dimension, consider a sequence $\theta = \theta_d$ with $\tau := \theta_d\sqrt{d} = O(1)$. By the classical normal approximation for spherical marginals, $Z := \sqrt{d}\,T$ converges in distribution to $\mathcal{N}(0, 1)$, which implies

$$q = \Pr(T < \theta_d) \to \Phi(\tau), \qquad 1 - q \to Q(\tau) := 1 - \Phi(\tau),$$

and

$$\alpha = \mathbb{E}[T \mid T \geq \theta_d] = \frac{1}{\sqrt{d}}\,\mathbb{E}[Z \mid Z \geq \tau]\,(1 + o(1)) = \frac{1}{\sqrt{d}}\,\lambda(\tau)\,(1 + o(1)),$$

where $\lambda(\tau) := \phi(\tau)/Q(\tau)$ and $\phi, \Phi$ are the standard normal pdf and cdf. Plugging these into $C(\theta, d)$ yields

$$C(\theta_d, d) = \frac{\Phi(\tau)}{Q(\tau)} \cdot \frac{1}{\lambda(\tau)^2} (1 + o(1)) = \frac{\Phi(\tau) Q(\tau)}{\phi(\tau)^2} (1 + o(1)) =: C(\tau) (1 + o(1)).$$

It remains to minimize $C(\tau) = \Phi(\tau)Q(\tau)/\phi(\tau)^2$ over $\tau \in \mathbb{R}$. The function $C$ is even since $\Phi(-\tau) = Q(\tau)$ and $\phi(-\tau) = \phi(\tau)$. Moreover,

$$C(0) = \frac{(1/2)(1/2)}{(1/\sqrt{2\pi})^2} = \frac{\pi}{2}.$$

A direct differentiation of $\log C(\tau)$ shows that $\tau = 0$ is the unique global minimizer:

$$\frac{d}{d\tau} \log C(\tau) = \frac{\phi(\tau)}{\Phi(\tau)} - \frac{\phi(\tau)}{Q(\tau)} + 2\tau,$$

which is strictly positive for $\tau > 0$ and strictly negative for $\tau < 0$. Consequently,

$$\inf_{\tau \in \mathbb{R}} C(\tau) = C(0) = \frac{\pi}{2}.$$

Therefore,

$$\liminf_{d \to \infty} \inf_{\theta \in [-1,1]} C(\theta, d) \geq \inf_{\tau \in \mathbb{R}} C(\tau) = \frac{\pi}{2},$$

which completes the proof. $\qquad\square$

## A.5. Proposition 3.8

**Proposition 3.8.** *Consider the input domain $\mathcal{X} = \mathbb{S}^0 \subset \mathbb{R}$ for $\mathrm{RR}(p)$. Let $\mathrm{RR}(p)$ be the randomized response local randomizer, and let $\chi_{\mathrm{lo}}$ be its lower shuffle index. Let $\widehat{x}_1$ be an unbiased estimator for $\mathrm{RR}(p)$. Then, $\chi_{\mathrm{lo}} \to \infty$,*

$$\mathsf{Err}_1(\mathrm{RR}(p), \widehat{x}_1) = \chi_{\mathrm{lo}}^2 \left(1 + \chi_{\mathrm{lo}}^{-1} + O(\chi_{\mathrm{lo}}^{-2})\right).$$

*Proof.* Let $d = 1$ and $\mathcal{X} = \mathbb{S}^0 = \{-1, +1\}$. For any $\theta \in (-1, 1)$, the spherical cap $\{u \in \mathbb{S}^0 : \langle u, x \rangle \geq \theta\}$ degenerates: since $\mathbb{S}^0 = \{\pm 1\}$ and $\langle u, x \rangle \in \{\pm 1\}$, we have $\langle u, x \rangle \geq \theta \iff \langle u, x \rangle = +1 \iff u = x$. Hence, for any such $\theta$, $\mathrm{PrivUnit}(p, \theta)$ is equivalent to the sign-flip (randomized response) mechanism

$$Y = \begin{cases} x & \text{with prob. } p, \\ -x & \text{with prob. } 1 - p, \end{cases} \qquad x \in \{\pm 1\}.$$

In particular, we may fix $\theta = 0$ without loss of generality and write $\mathrm{RR}(p)$ for this mechanism.

**Unbiased estimation and $\mathsf{Err}_1$.** Let $t := 2p - 1 \in (0, 1)$ (we assume $p \geq 1/2$; this is the relevant high privacy direction). Then $\mathbb{E}[Y \mid x] = tx$. The estimator $\widehat{x}_1(Y) := Y/t$ is unbiased since $\mathbb{E}[\widehat{x}_1(Y) \mid x] = \mathbb{E}[Y \mid x]/t = x$. Moreover, since $Y^2 = 1$ a.s.,

$$\mathsf{Err}_1(\mathrm{RR}(p), \widehat{x}_1) = \sup_{x \in \{\pm 1\}} \mathbb{E}\left[\left(\frac{Y}{t} - x\right)^2 \Big| x\right] = \sup_x \left(\frac{1}{t^2}\mathbb{E}[Y^2 \mid x] - 2\frac{1}{t}\mathbb{E}[xY \mid x] + 1\right) = \frac{1}{t^2} - 1.$$

**Computing $\chi_{\mathrm{lo}}$.** Let $\mathcal{R}_x$ denote the law of $Y$ given input $x \in \{\pm 1\}$. Under the zero-out convention we take $\mathcal{R}_\perp = \mathcal{R}_{\mathrm{BG}}$. The maximal blanket mass is

$$\gamma = \sum_{y \in \{\pm 1\}} \inf_{x \in \{\pm 1\}} \mathcal{R}_x(y).$$

When $p \geq 1/2$, we have $\inf_x \mathcal{R}_x(+1) = 1 - p$ and $\inf_x \mathcal{R}_x(-1) = 1 - p$, hence

$$\gamma = 2(1 - p) = 1 - t, \qquad \text{and} \qquad \mathcal{R}_{\mathrm{BG}}(+1) = \mathcal{R}_{\mathrm{BG}}(-1) = \tfrac{1}{2}.$$

For neighboring inputs $(x, \perp)$, the generalized amplification variable at $\varepsilon = 0$ is

$$\ell_0(y; x, \perp, \mathcal{R}_{\mathrm{BG}}) = \frac{\mathcal{R}_x(y) - \mathcal{R}_{\mathrm{BG}}(y)}{\mathcal{R}_{\mathrm{BG}}(y)}.$$

Taking $x = +1$ (the case $x = -1$ is identical), we have $\mathcal{R}_{+1}(+1) = p = (1 + t)/2$ and $\mathcal{R}_{+1}(-1) = 1 - p = (1 - t)/2$, so

$$\ell_0(+1) = t, \qquad \ell_0(-1) = -t.$$

Therefore, for $Y \sim \mathcal{R}_{\mathrm{BG}}$, $\mathrm{Var}[\ell_0(Y)] = t^2$, and by the definition of $\chi_{\mathrm{lo}}$,

$$\chi_{\mathrm{lo}}^2 = \frac{1}{\frac{1}{\gamma}\mathrm{Var}_{Y \sim \mathcal{R}_{\mathrm{BG}}}[\ell_0(Y)]} = \frac{\gamma}{t^2} = \frac{1 - t}{t^2}.$$

Equivalently, $t$ solves the quadratic equation

$$\chi_{\mathrm{lo}}^2 t^2 + t - 1 = 0,$$

so (taking the positive root)

$$t = \frac{-1 + \sqrt{1 + 4\chi_{\mathrm{lo}}^2}}{2\chi_{\mathrm{lo}}^2}. \tag{37}$$

**Expansion for $\chi_{\mathrm{lo}} \to \infty$.** Using $\sqrt{1 + 4\chi_{\mathrm{lo}}^2} = 2\chi_{\mathrm{lo}}\sqrt{1 + \frac{1}{4\chi_{\mathrm{lo}}^2}} = 2\chi_{\mathrm{lo}}\left(1 + \frac{1}{8\chi_{\mathrm{lo}}^2} + O(\chi_{\mathrm{lo}}^{-4})\right)$, (37) yields

$$t = \frac{1}{\chi_{\mathrm{lo}}} - \frac{1}{2\chi_{\mathrm{lo}}^2} + O(\chi_{\mathrm{lo}}^{-3}).$$

Hence,

$$\frac{1}{t^2} = \chi_{\mathrm{lo}}^2 \left(1 + \frac{1}{\chi_{\mathrm{lo}}} + O(\chi_{\mathrm{lo}}^{-2})\right),$$

and therefore

$$\mathsf{Err}_1(\mathrm{RR}(p), \widehat{x}_1) = \left(\frac{1}{t^2}\right) - 1 = \chi_{\mathrm{lo}}^2\left(1 + \chi_{\mathrm{lo}}^{-1} + O(\chi_{\mathrm{lo}}^{-2})\right), \qquad (\chi_{\mathrm{lo}} \to \infty),$$

where the subtraction of 1 is absorbed into the $O(\chi_{\mathrm{lo}}^{-2})$ term after factoring out $\chi_{\mathrm{lo}}^2$. This proves the claimed expansion. $\qquad\square$

### A.6. Theorem 4.1

**Theorem 4.1.** *Fix $d \geq 1$. There exists a choice of parameters $(\gamma, \sigma_0)$ such that $\chi_{\mathrm{lo}}(\mathcal{R}^{\mathsf{BMG}}) \geq \chi$ and, as $\chi \to \infty$,*

$$\mathsf{Err}_1(\mathcal{R}^{\mathsf{BMG}}, \widehat{x}) = d\chi^2\left(1 + \frac{3}{2}\chi^{-2/3} + O\left(\chi^{-4/3}\right)\right).$$

*Proof.* Throughout, write $A := 1 - \gamma \in (0, 1]$ and recall that the blanket distribution is $\mathcal{R}_{\mathrm{BG}} = \mathcal{N}(0, \sigma_0^2 I_d)$.

**Step 1: Exact formula for $\chi_{\mathrm{lo}}(\mathcal{R})$.** Fix $x \in \mathcal{X}$ and let $\phi_{\sigma_0}(y)$ denote the density of $\mathcal{N}(0, \sigma_0^2 I_d)$. The released distribution under input $x$ is the mixture

$$\mathcal{R}_x = \gamma \mathcal{N}(0, \sigma_0^2 I_d) + A\mathcal{N}(x, \sigma_0^2 I_d), \qquad \mathcal{R}_\perp = \mathcal{R}_{\mathrm{BG}} = \mathcal{N}(0, \sigma_0^2 I_d).$$

Hence, with respect to $\mathcal{R}_{\mathrm{BG}}$, the likelihood ratio is

$$\frac{d\mathcal{R}_x}{d\mathcal{R}_{\mathrm{BG}}}(y) = \gamma + A\frac{\phi_{\sigma_0}(y - x)}{\phi_{\sigma_0}(y)} = \gamma + A\exp\left(\frac{\langle x, y \rangle}{\sigma_0^2} - \frac{\|x\|_2^2}{2\sigma_0^2}\right).$$

For the zero-out neighbor pair $(x, \perp)$, the corresponding centered blanket-referenced difference equals

$$\ell_0(y; x, \perp, \mathcal{R}_{\mathrm{BG}}) = \frac{\mathcal{R}_x(y) - \mathcal{R}_\perp(y)}{\mathcal{R}_{\mathrm{BG}}(y)} = \frac{d\mathcal{R}_x}{d\mathcal{R}_{\mathrm{BG}}}(y) - 1 = A\left(\exp\left(\frac{\langle x, y \rangle}{\sigma_0^2} - \frac{\|x\|_2^2}{2\sigma_0^2}\right) - 1\right).$$

Under $Y \sim \mathcal{R}_{\mathrm{BG}} = \mathcal{N}(0, \sigma_0^2 I_d)$ we have $\langle x, Y \rangle \sim \mathcal{N}(0, \sigma_0^2 \|x\|_2^2)$, and therefore

$$\mathbb{E}\Big[\exp\Big(\frac{\langle x, Y \rangle}{\sigma_0^2} - \frac{\|x\|_2^2}{2\sigma_0^2}\Big)\Big] = 1, \qquad \mathbb{E}\Big[\exp\Big(2\frac{\langle x, Y \rangle}{\sigma_0^2} - \frac{\|x\|_2^2}{\sigma_0^2}\Big)\Big] = \exp\Big(\frac{\|x\|_2^2}{\sigma_0^2}\Big).$$

It follows that

$$\mathrm{Var}_{Y \sim \mathcal{R}_{\mathrm{BG}}}[\ell_0(Y; x, \perp, \mathcal{R}_{\mathrm{BG}})] = A^2 \Big(\exp(\|x\|_2^2/\sigma_0^2) - 1\Big).$$

By monotonicity in $\|x\|_2$ and $\sup_{\|x\|_2 \le 1} \|x\|_2 = 1$, the worst case occurs at $\|x\|_2 = 1$, hence

$$\sup_{x \simeq x'} \sqrt{\frac{1}{\gamma} \mathrm{Var}_{Y \sim \mathcal{R}_{\mathrm{BG}}}[\ell_0(Y; x, x', \mathcal{R}_{\mathrm{BG}})]} = \sqrt{\frac{A^2}{\gamma}\Big(e^{1/\sigma_0^2} - 1\Big)}.$$

Recalling the definition of $\chi_{\mathrm{lo}}$, we obtain the exact identity

$$\chi_{\mathrm{lo}}(\mathcal{R})^2 = \frac{\gamma}{A^2} \cdot \frac{1}{e^{1/\sigma_0^2} - 1} = \frac{\gamma}{(1-\gamma)^2} \cdot \frac{1}{e^{1/\sigma_0^2} - 1}.$$

**Step 2: Worst-case squared error of the unbiased estimator.** Consider the linear estimator $\widehat{x}(Y) = Y/A$, which is unbiased. For any $x \in \mathcal{X}$, since $\widehat{x}(Y)$ is unbiased for $x$,

$$\mathbb{E}\big[\|\widehat{x}(Y) - x\|_2^2\big] = \mathrm{tr}(\mathrm{Cov}(\widehat{x}(Y))).$$

The distribution $\mathcal{R}_x$ is a two-component Gaussian mixture with common covariance $\sigma_0^2 I_d$ and means $0$ and $x$. Thus

$$\mathbb{E}[Y] = (1-\gamma)x = Ax, \qquad \mathrm{Cov}(Y) = \sigma_0^2 I_d + \gamma A \, xx^\top.$$

Here,

$$\mathrm{Cov}(Y) = \mathbb{E}[\mathrm{Cov}(Y \mid B)] + \mathrm{Cov}(\mathbb{E}[Y \mid B]).$$

Since $\mathrm{Cov}(Y \mid B) = \sigma_0^2 I_d$ for both $B = 0, 1$,

$$\mathbb{E}[\mathrm{Cov}(Y \mid B)] = \sigma_0^2 I_d.$$

and

$$\mathrm{Cov}(\mathbb{E}[Y \mid B]) = \gamma(0 - Ax)(0 - Ax)^\top + A(x - Ax)(x - Ax)^\top = \gamma A^2 xx^\top + A\gamma^2 xx^\top = \gamma A \, xx^\top.$$

Therefore,

$$\mathrm{Cov}(Y) = \sigma_0^2 I_d + \gamma A \, xx^\top.$$

Therefore,

$$\mathrm{Cov}(\widehat{x}(Y)) = \frac{1}{A^2} \mathrm{Cov}(Y) = \frac{\sigma_0^2}{A^2} I_d + \frac{\gamma}{A} xx^\top,$$

and taking the trace yields the exact squared error

$$\mathbb{E}\big[\|\widehat{x}(Y) - x\|_2^2\big] = \frac{d \, \sigma_0^2}{A^2} + \frac{\gamma}{A}\|x\|_2^2$$

Maximizing over $\|x\|_2 \le 1$ gives the worst-case single-user MSE

$$\sup_{x \in \mathcal{X}} \mathbb{E}\big[\|\widehat{x}(Y) - x\|_2^2\big] = \frac{d \, \sigma_0^2}{A^2} + \frac{\gamma}{A}. \tag{38}$$

**Step 3: Tuning of $(\gamma, \sigma_0)$ under $\chi_{\mathrm{lo}}(\mathcal{R}) \geq \chi$.** Fix $\chi > 0$ and write $A := 1 - \gamma \in (0, 1)$ and $t := 1/\sigma_0^2 > 0$. Imposing the equality constraint $\chi_{\mathrm{lo}}(\mathcal{R}) = \chi$ is without loss of generality for an upper bound (we are free to pick any feasible parameters), and yields the exact relation

$$e^t - 1 = \frac{\gamma}{A^2 \chi^2} = \frac{1 - A}{A^2 \chi^2} \qquad \Longleftrightarrow \qquad t = \log\left(1 + \frac{1 - A}{A^2 \chi^2}\right). \tag{39}$$

Define

$$u := \frac{1 - A}{A^2 \chi^2}.$$

Then $t = \log(1 + u)$. In the strong-privacy regime we will choose $A \to 0$, hence $u \to 0$ as $\chi \to \infty$.

**Choice of parameters.** Set

$$A := \chi^{-2/3}, \qquad \gamma := 1 - A, \qquad t := \log(1 + u), \qquad \sigma_0^2 := 1/t, \tag{40}$$

where $u = (1 - A)/(A^2 \chi^2)$ as above. With this choice,

$$u = \frac{1 - A}{A^2 \chi^2} = \frac{1}{A^2 \chi^2} - \frac{1}{A \chi^2} = \chi^{-2/3} - \chi^{-4/3}. \tag{41}$$

In particular, for all sufficiently large $\chi$, we have $0 < u \leq 1/2$, so the Taylor expansion of $\log(1 + u)$ with a controlled remainder applies.

**A useful analytic bound.** For $u \in (0, 1/2]$, define

$$R_{\log}(u) := \log(1 + u) - \left(u - \frac{u^2}{2}\right).$$

A standard remainder bound (e.g. from the Lagrange form of the Taylor remainder, using that $|(\log(1 + u))^{(3)}| = 2/(1 + u)^3 \leq 2$ on $[0, 1/2]$) gives

$$|R_{\log}(u)| \leq C_{\log}\, u^3 \qquad \text{for all } u \in (0, 1/2], \tag{42}$$

for some universal constant $C_{\log} > 0$. Consequently,

$$t = \log(1 + u) = u\left(1 - \frac{u}{2} + \rho(u)\right), \qquad |\rho(u)| \leq C_{\log} u^2. \tag{43}$$

**Expansion of the dominant term $1/(A^2 t)$.** Using (43),

$$\frac{1}{A^2 t} = \frac{1}{A^2 u} \cdot \frac{1}{1 - \frac{u}{2} + \rho(u)}.$$

For $\chi$ large enough, $u$ is small and thus $\left|-\frac{u}{2} + \rho(u)\right| \leq 1/4$, so we may use the expansion $\frac{1}{1+z} = 1 - z + O(z^2)$ with $z = -\frac{u}{2} + \rho(u)$ to obtain

$$\frac{1}{1 - \frac{u}{2} + \rho(u)} = 1 + \frac{u}{2} + O(u^2), \tag{44}$$

where the $O(u^2)$ term is uniform for all sufficiently large $\chi$.

Next, observe that

$$\frac{1}{A^2 u} = \frac{1}{A^2} \cdot \frac{A^2 \chi^2}{1 - A} = \frac{\chi^2}{1 - A}. \tag{45}$$

Since $A = \chi^{-2/3} \to 0$, we also have the geometric expansion

$$\frac{1}{1 - A} = 1 + A + O(A^2). \tag{46}$$

Combining (44), (45), and (46) yields

$$\frac{1}{A^2 t} = \chi^2 \left(1 + A + \frac{u}{2} + O(A^2 + u^2 + Au)\right). \tag{47}$$

With our choice (40) and (41), we have $A = \chi^{-2/3}$ and $u = \chi^{-2/3} + O(\chi^{-4/3})$, so

$$A + \frac{u}{2} = \chi^{-2/3} + \frac{1}{2}\chi^{-2/3} + O(\chi^{-4/3}) = \frac{3}{2}\chi^{-2/3} + O(\chi^{-4/3}),$$

and moreover $A^2 + u^2 + Au = O(\chi^{-4/3})$. Plugging into (47) gives the refined asymptotic

$$\frac{1}{A^2 t} = \chi^2 \left(1 + \frac{3}{2}\chi^{-2/3} + O(\chi^{-4/3})\right). \tag{48}$$

**Mixture penalty term.** The second term in the exact worst-case MSE (38) is

$$\frac{\gamma}{A} = \frac{1-A}{A} = \frac{1}{A} - 1 = \chi^{2/3} - 1.$$

Therefore, for fixed $d \geq 1$,

$$\frac{\gamma/A}{d\chi^2} = O(\chi^{-4/3}), \tag{49}$$

so it contributes only to the $O(\chi^{-4/3})$ remainder in the relative error.

**Conclusion.** Substituting (48) and (49) into the exact MSE expression (38) yields

$$\sup_{\|x\|_2 \leq 1} \mathbb{E}\left[\|\widehat{x}(Y) - x\|_2^2\right] = d\chi^2 \left(1 + \frac{3}{2}\chi^{-2/3} + O(\chi^{-4/3})\right),$$

with $(\gamma, \sigma_0)$ as in (40). This proves the claimed upper bound for $\mathsf{Err}_1$ under the constraint $\chi_{\mathrm{lo}}(\mathcal{R}) \geq \chi$.

$\square$

### A.7. Theorem 3.3

**Theorem 3.3** (Gaussian Limit Correspondence). *Fix a constant $\sigma > 0$. In the $\sigma$-high privacy regime $\{(\varepsilon_n, \delta_n)\}_{n \geq 1}$*

$$d\sigma^2 \leq \mathsf{Err}^\star_{\mathrm{DP}}(\{(\varepsilon_n, \delta_n)\}_{n \geq 1}) \leq d\sigma^2 \left(1 + \eta(\sigma)\right),$$

*where $\eta(\sigma) \geq 0$ depends only on $\sigma$ and satisfies $\eta(\sigma) = O(\sigma^{-2/3})$ as $\sigma \to \infty$.*

*Proof.* Fix $\sigma > 0$ and assume that $\{(\varepsilon_n, \delta_n)\}_{n \geq 1}$ lies in the $\sigma$-high privacy regime (Definition 3.1).

**Step 1: Reduction to shuffle-index constrained problems.** By Lemma 3.4, for any $\eta > 0$,

$$\mathsf{Err}^\star_{\mathrm{up}}(\sigma) \leq \mathsf{Err}^\star_{\mathrm{DP}}(\{(\varepsilon_n, \delta_n)\}_{n \geq 1}) \leq \mathsf{Err}^\star_{\mathrm{lo}}(\sigma + \eta). \tag{50}$$

**Step 2: Lower bound.** Consider the single-user optimization problem (6) with constraint $\chi_{\mathrm{up}}(\mathcal{R}) \geq \sigma$. By Theorem 3.5, for any feasible pair $(\mathcal{R}, \widehat{x})$ we have

$$\mathsf{Err}_1(\mathcal{R}, \widehat{x}) \geq d\chi_{\mathrm{up}}(\mathcal{R})^2 \geq d\sigma^2.$$

Taking the infimum over all feasible $(\mathcal{R}, \widehat{x})$ yields

$$\mathsf{Err}^\star_{\mathrm{up}}(\sigma) \geq d\sigma^2. \tag{51}$$

Combining (50) and (51) gives

$$\mathsf{Err}^\star_{\mathrm{DP}}(\{(\varepsilon_n, \delta_n)\}_{n \geq 1}) \geq d\sigma^2.$$

**Step 3: Upper bound.** Fix $\eta = 1$. We invoke the explicit upper bound proved in Corollary 3.6: there exist constants $\chi_0 \geq 1$ and $C > 0$ such that for all $\chi \geq \chi_0$,

$$\mathsf{Err}^\star_{\mathrm{lo}}(\chi) \leq d\chi^2 \left(1 + \frac{3}{2}\chi^{-2/3} + C\chi^{-4/3}\right). \tag{52}$$

Using the right inequality in (50) (with $\eta = 1$) yields

$$\mathsf{Err}^\star_{\mathrm{DP}}(\{(\varepsilon_n, \delta_n)\}_{n\geq1}) \leq \mathsf{Err}^\star_{\mathrm{lo}}(\sigma + 1). \tag{53}$$

**Case 1:** $\sigma + 1 \geq \chi_0$**.** Combining (53) with (52) (applied at $\chi = \sigma + 1$) gives

$$\mathsf{Err}^\star_{\mathrm{DP}}(\{(\varepsilon_n, \delta_n)\}_{n\geq1}) \leq d(\sigma + 1)^2 \left(1 + \frac{3}{2}(\sigma + 1)^{-2/3} + C(\sigma + 1)^{-4/3}\right). \tag{54}$$

Define for $\sigma \geq \chi_0 - 1$,

$$\eta(\sigma) := \frac{(\sigma + 1)^2}{\sigma^2} \left(1 + \frac{3}{2}(\sigma + 1)^{-2/3} + C(\sigma + 1)^{-4/3}\right) - 1. \tag{55}$$

Then (54) rewrites as

$$\mathsf{Err}^\star_{\mathrm{DP}}(\{(\varepsilon_n, \delta_n)\}_{n\geq1}) \leq d\sigma^2(1 + \eta(\sigma)), \qquad \text{for all } n \text{ sufficiently large.}$$

Moreover, since the bracketed factor in (55) is at least 1 and $(\sigma + 1)^2/\sigma^2 \geq 1$, we have $\eta(\sigma) \geq 0$ for all $\sigma \geq \chi_0 - 1$. Finally, as $\sigma \to \infty$,

$$\eta(\sigma) = \left(\frac{(\sigma + 1)^2}{\sigma^2} - 1\right) + \frac{(\sigma + 1)^2}{\sigma^2}\left(\frac{3}{2}(\sigma + 1)^{-2/3} + C(\sigma + 1)^{-4/3}\right) = O(\sigma^{-1}) + O(\sigma^{-2/3}) = O(\sigma^{-2/3}).$$

**Case 2:** $\sigma + 1 < \chi_0$**.** In this case, we simply define $\eta(\sigma) \geq 0$ large enough so that

$$\mathsf{Err}^\star_{\mathrm{lo}}(\sigma + 1) \leq d\sigma^2(1 + \eta(\sigma)).$$

For example, it suffices to take

$$\eta(\sigma) := \max\left\{0, \frac{\mathsf{Err}^\star_{\mathrm{lo}}(\sigma + 1)}{d\sigma^2} - 1\right\}.$$

With this definition, the bound

$$\mathsf{Err}^\star_{\mathrm{DP}}(\{(\varepsilon_n, \delta_n)\}_{n\geq1}) \leq d\sigma^2(1 + \eta(\sigma))$$

holds for all $n$ sufficiently large by (53). Moreover, this definition affects $\eta(\sigma)$ only on the bounded interval $\sigma \in (0, \chi_0 - 1)$ and therefore does not change the asymptotic statement $\eta(\sigma) = O(\sigma^{-2/3})$ as $\sigma \to \infty$.

**Conclusion.** Combining the lower and upper bounds established above completes the proof. $\square$

### A.8. Proposition 4.2

**Proposition 4.2.** *Consider the Gaussian local randomizer* $\mathcal{R}^{\mathsf{GL}}_x = \mathcal{N}(x, \sigma_0^2 I_d), x \in \mathbb{B}_2^d$, *together with the unbiased estimator* $\widehat{x}(Y) = Y$. *Let* $\chi_{\mathrm{chua}} := 1/\sqrt{\mathrm{Var}_{Y \sim \mathcal{R}^{\mathsf{GL}}_{-e}}[\ell_0(Y; e, 0, \mathcal{R}^{\mathsf{GL}}_{-e})]}$ *where* $e \in \mathbb{S}^{d-1}$. *Then, as* $\sigma_0 \to \infty$ *(equivalently* $\chi_{\mathrm{chua}} \to \infty$*),*

$$\mathsf{Err}_1(\mathcal{R}^{\mathsf{GL}}, \widehat{x}) = d\chi^2_{\mathrm{chua}} \left(1 + \frac{9}{2}\chi^{-2}_{\mathrm{chua}} + O(\chi^{-4}_{\mathrm{chua}})\right).$$

*Proof.* Consider the Gaussian local randomizer

$$\mathcal{R}_x = \mathcal{N}(x, \sigma_0^2 I_d), \qquad x \in \mathbb{B}_2^d,$$

which corresponds to the case $\gamma = 0$. A natural unbiased per-message estimator is

$$\widehat{x}(Y) = Y, \qquad \text{since } \mathbb{E}[Y \mid x] = x.$$

Hence the (single-user) worst-case squared error is

$$\mathsf{Err}_1(\mathcal{R}, \widehat{x}) := \sup_{x \in \mathbb{B}_2^d} \mathbb{E}\big[\|\widehat{x}(Y) - x\|_2^2\big] = \mathbb{E}\|Z\|_2^2 = d\sigma_0^2, \qquad Z \sim \mathcal{N}(0, \sigma_0^2 I_d). \tag{56}$$

**Step 1: computing a concrete variance term for $\chi_{\text{chua}}$.** Recall

$$\ell_0(y; x_1, x_1', \mathcal{R}_x) = \frac{\mathcal{R}_{x_1}(y) - \mathcal{R}_{x_1'}(y)}{\mathcal{R}_x(y)}.$$

Write $\sigma^2 := \sigma_0^2$ and $\mathcal{R}_\mu = \mathcal{N}(\mu, \sigma^2)$.

Let $U$ be any orthogonal matrix. Since $Y \sim \mathcal{N}(\mu, \sigma^2 I_d)$ implies $UY \sim \mathcal{N}(U\mu, \sigma^2 I_d)$, and Gaussian density ratios are preserved under the change of variables $y \mapsto Uy$, we have

$$\ell_0(Y; x_1, x_1', \mathcal{R}_x) \stackrel{d}{=} \ell_0(UY; Ux_1, Ux_1', \mathcal{R}_{Ux}),$$

and hence $\mathrm{Var}[\ell_0(Y; x_1, x_1', \mathcal{R}_x)]$ is invariant under applying the same rotation to $(x_1, x_1', x)$. Therefore, we may rotate coordinates so that $x_1 = e_1$, $x_1' = 0$, and $x = -e_1$. Moreover, $\ell_0(y; e_1, 0, \mathcal{R}_{-e_1})$ depends on $y$ only through the one-dimensional projection $\langle y, e_1 \rangle$, and for $Y \sim \mathcal{N}(-e_1, \sigma^2 I_d)$ we have $\langle Y, e_1 \rangle \sim \mathcal{N}(-1, \sigma^2)$. Identifying this coordinate with $\mathbb{R}$, we may write $x_1 = 1$, $x_1' = 0$, and $x = -1$ without loss of generality.

Using the Gaussian density ratio identity

$$\frac{\phi_{\mu_1, \sigma^2}(y)}{\phi_{\mu_2, \sigma^2}(y)} = \exp\left(\frac{(\mu_1 - \mu_2)y}{\sigma^2} - \frac{\mu_1^2 - \mu_2^2}{2\sigma^2}\right),$$

we obtain

$$\frac{\mathcal{R}_1(y)}{\mathcal{R}_{-1}(y)} = \exp\left(\frac{2y}{\sigma^2}\right), \qquad \frac{\mathcal{R}_0(y)}{\mathcal{R}_{-1}(y)} = \exp\left(\frac{y}{\sigma^2} + \frac{1}{2\sigma^2}\right),$$

and therefore

$$\ell_0(y) = \ell_0(y; 1, 0, \mathcal{R}_{-1}) = \exp\left(\frac{2y}{\sigma^2}\right) - \exp\left(\frac{y}{\sigma^2} + \frac{1}{2\sigma^2}\right). \tag{57}$$

Let $Y \sim \mathcal{R}_x = \mathcal{N}(-1, \sigma^2)$. Using the Gaussian MGF $\mathbb{E}[e^{tY}] = \exp(t\mu + \frac{1}{2}t^2\sigma^2)$ with $\mu = -1$, one checks $\mathbb{E}[\ell_0(Y)] = 0$, and hence $\mathrm{Var}[\ell_0(Y)] = \mathbb{E}[\ell_0(Y)^2]$. Expanding the square in (57) and taking expectations term-by-term yields

$$\mathrm{Var}_{Y \sim \mathcal{N}(-1, \sigma^2)}[\ell_0(Y)] = e^{4/\sigma^2} - 2e^{2/\sigma^2} + e^{1/\sigma^2}. \tag{58}$$

$$V(\sigma^2) := e^{4/\sigma^2} - 2e^{2/\sigma^2} + e^{1/\sigma^2},$$

Then, by definition of $\chi_{\text{chua}}$,

$$\chi_{\text{chua}}(\mathcal{R})^2 = \frac{1}{\mathrm{Var}_{Y \sim \mathcal{R}_x}[\ell_0(Y; x_1, x_1', \mathcal{R}_x)]} = \frac{1}{V(\sigma^2)}.$$

**Step 2: asymptotic expansion as $\sigma_0 \to \infty$.** Let $a := 1/\sigma^2$. Using $e^{ca} = 1 + ca + \frac{c^2}{2}a^2 + \frac{c^3}{6}a^3 + \frac{c^4}{24}a^4 + O(a^5)$, we expand (58):

$$V(\sigma^2) = a + \frac{9}{2}a^2 + \frac{49}{6}a^3 + \frac{75}{8}a^4 + O(a^5). \tag{59}$$

Inverting the series gives

$$\chi_{\text{chua}}^2(\sigma^2) = \frac{1}{V(\sigma^2)} = \frac{1}{a}\left(1 - \frac{9}{2}a + \frac{145}{12}a^2 - 27a^3 + O(a^4)\right)$$

$$= \sigma^2 - \frac{9}{2} + \frac{145}{12}\frac{1}{\sigma^2} - \frac{27}{\sigma^4} + O\left(\frac{1}{\sigma^6}\right). \tag{60}$$

**Step 3: expressing $\sigma_0^2$ (hence $\mathsf{Err}_1$) as a series in $\chi_{\text{chua}}$.** Reverting the expansion (60) yields, as $\chi_{\text{chua}} \to \infty$,

$$\sigma^2 = \chi_{\text{chua}}^2 + \frac{9}{2} - \frac{145}{12}\chi_{\text{chua}}^{-2} + \frac{651}{8}\chi_{\text{chua}}^{-4} + O(\chi_{\text{chua}}^{-6}). \tag{61}$$

**Conclusion (error expansion).** Combining (56) and (61), we obtain

$$\mathsf{Err}_1(\mathcal{R}, \widehat{x}) = d\,\sigma_0^2 = d\,\chi_{\text{chua}}^2 + \frac{9}{2}d - \frac{145}{12}d\,\chi_{\text{chua}}^{-2} + O\big(d\,\chi_{\text{chua}}^{-4}\big)\,, \qquad (\sigma_0 \to \infty).$$

$\square$

## B. Regularity conditions for the local randomizer

**Assumption B.1** (Regularity conditions for the local randomizer)**.** Consider the local randomizer $\mathcal{R} : \mathcal{X}_\perp \to \mathcal{Y}$ and recall that we set $\mathcal{R}_\perp := \mathcal{R}_{\text{BG}}$ under zero-out adjacency. We assume that there exists $\rho_0 > 0$ such that, for every pair $x_1 \neq x_1' \in \mathcal{X}_\perp$ and every reference distribution $\mathcal{R}_{\text{ref}} \in \{\mathcal{R}_{\text{BG}}\} \cup \{\mathcal{R}_x : x \in \mathcal{X}\}$, the following conditions hold uniformly over all $\varepsilon \leq \rho_0$.

(1) **Uniform moment bounds.** For every integer $k \geq 1$, $\mathbb{E}_{Y \sim \mathcal{R}_{\text{ref}}}\big[\,|\ell_\varepsilon(Y; x_1, x_1', \mathcal{R}_{\text{ref}})|^k\,\big] < \infty$.

(2) **Non-degenerate variance.** The variance $\sigma^2 := \mathrm{Var}_{Y \sim \mathcal{R}_{\text{ref}}}\big(\ell_0(Y; x_1, x_1', \mathcal{R}_{\text{ref}})\big)$ is strictly positive when $\mathcal{R}_{x_1} \neq \mathcal{R}_{x_1'}$. Moreover, the following conditions hold: $\int \frac{\mathcal{R}_{x_1}(y)^2}{\mathcal{R}_{\text{ref}}(y)}\,dy < \infty$ and $\int \frac{\mathcal{R}_{x_1'}(y)^2}{\mathcal{R}_{\text{ref}}(y)}\,dy < \infty$.

(3) **Structural condition.** Either of the following holds.

(Cont) $\ell_\varepsilon(Y; x_1, x_1', \mathcal{R}_{\text{ref}})$ has a nontrivial absolutely continuous component with respect to Lebesgue measure whose support contains a fixed nondegenerate bounded interval $I \subset \mathbb{R}$ independent of $\varepsilon$.

(Bound) There exists a constant $C < \infty$ such that $|\ell_\varepsilon(Y; x_1, x_1', \mathcal{R}_{\text{ref}})| \leq C$ almost surely.

Assumption B.1 is mild: any nontrivial local randomizer that satisfies pure LDP automatically satisfies these conditions, and even when pure LDP does not hold, if the privacy amplification random variable $\ell_0(Y; x_1, x_1', \mathcal{R}_{\text{ref}})$ is non-degenerate (i.e., not almost surely constant), then one can enforce the bounded case (Bound) in (3) by truncating the output space to a large but bounded interval (i.e., by slightly modifying the local randomizer).

## C. Computing the shuffled $(\varepsilon, \delta)$ guarantee

Lemma 5.3 of Balle et al. (Balle et al., 2019) implies that the shuffled blanket-mixed Gaussian mechanism is $(\varepsilon, \delta(\varepsilon))$-DP with

$$\delta(\varepsilon) \;\leq\; \frac{1}{n\gamma}\, \mathbb{E}\!\left[\left(\sum_{i=1}^{M_0} \ell_\varepsilon(Y_i)\right)_{\!+}\right]$$

where

$$M_0 \sim \mathrm{Bin}(n, \gamma), \;\; Y_1, Y_2, \ldots \overset{\text{i.i.d.}}{\sim} \mathcal{R}_{\text{BG}}^{\text{BMG}},$$

and $(a)_+ := \max\{a, 0\}$.

In the zero-out adjacency model, the neighboring inputs are $x \in \mathcal{X}$ and $\perp$, and $\mathcal{R}_\perp^{\text{BMG}}$ is the blanket distribution.

$$\mathcal{R}_{\text{BG}}^{\text{BMG}} = \mathcal{R}_\perp^{\text{BMG}} = \mathcal{N}(0, \sigma_0^2 I_d),$$
$$\mathcal{R}_x^{\text{BMG}} = \gamma\,\mathcal{N}(0, \sigma_0^2 I_d) + (1-\gamma)\,\mathcal{N}(x, \sigma_0^2 I_d).$$

Hence, for any $\varepsilon > 0$

$$\ell_\varepsilon(y) := \frac{\mathcal{R}_x^{\text{BMG}}(y) - e^\varepsilon \mathcal{R}_\perp^{\text{BMG}}(y)}{\mathcal{R}_{\text{BG}}^{\text{BMG}}(y)}$$
$$= \gamma + (1-\gamma)\exp\!\left(\frac{\langle y, x\rangle}{\sigma_0^2} - \frac{\|x\|_2^2}{2\sigma_0^2}\right) - e^\varepsilon. \tag{62}$$

Therefore, we can compute the distribution of $\ell_\varepsilon(Y)$, which enables the rigorous numerical upper bounds using the FFT-based accountant of Takagi & Liew (2026), which computes the required convolution terms efficiently with explicit truncation/discretization/wrap-around error bounds.

