# OpenReview forum: "Shuffling-Aware Optimization for Private Vector Mean Estimation"
_ICML.cc/2026/Conference — ICML 2026 regular_

### Official Review · Reviewer_APgj · 2026-03-11

**Soundness:** 3
**Presentation:** 2
**Significance:** 3
**Originality:** 3
**Overall Recommendation:** 4
**Confidence:** 2

**Summary:**

This paper studies unbiased estimation of a d-dimensional mean under shuffle differential privacy. The authors analyze the problem through the notion of the shuffle index, which allows them to derive lower bounds against unbiased mechanisms. They then introduce a new mechanism based on a mixture of a Gaussian mechanism and a zero-centered Gaussian component. Interestingly, this mechanism matches the lower bound in the high-privacy regime, whereas the standard Gaussian mechanism does not.

**Compliance With Llm Reviewing Policy:**

Affirmed.

**Final Justification:**

The authors have answered my questions in their rebuttal. While I may have a doubt about the fact of only comparing to unbiased mechanisms, I understand that the article is of interest. I thus believe that my original "weak accept" recommendation is appropriate.

**Key Questions For Authors:**

- Was the idea of introducing a “blanket distribution” already known before this work?

- Could the suboptimality of the standard Gaussian mechanism in the high-privacy regime, compared with the version augmented by a blanket, suggest that under strong privacy constraints, methods such as DP-SGD might also be improved using similar ideas?

- In Equation (1), why is the noise added at scale $1/\sqrt{n}$? Since the sensitivity scales as $1/n$, one might expect the noise to scale as 1/n. Is this because \varepsilon scales as $A/\sqrt{n}$? A brief remark would help.

- If I understand correctly, the role of the blanket is that a sample drawn from it is independent of the original input, and therefore does not reveal information about the original data point even after permutation. If so, this intuition could be introduced much earlier in the paper, as it is very helpful for understanding the construction.

- Is the unbiasedness requirement imposed on the aggregated estimator only, or also on each individual user contribution?

- In Proposition 4.2, what is the quantification over $e$?

- Much of the analysis seems to rely on Lemma 2.6, which states that chi is a good proxy for the privacy profile and uses asymptotic quantities in its result. Later, however, the paper also presents asymptotic results in which the Chi’s vary. How is this double limiting regime handled mathematically?

**Limitations:**

Yes

**Strengths And Weaknesses:**

Strengths

- The role of shuffling in differential privacy is likely one of the areas in private machine learning where the most interesting phenomena remain to be uncovered, and this paper aims to make progress in that direction

- The paper appears to introduce new tools for the differential privacy community, in particular the blanketed Gaussian mechanism and related mathematical techniques.

- The result showing that the proposed mechanism matches the lower bound in the high-privacy regime is theoretically interesting.

Weaknesses
- The results seem mostly interesting in the high-privacy regime, and I am not entirely sure this is the regime of greatest practical interest to the community. That said, the results are still theoretically valuable.

- As stated by the authors, Lemma 2.6 is only meaningful in the high-privacy regime. This does not make the result less interesting, but it does limit its applicability.

- The focus on unbiased estimators seems potentially restrictive. Usually, one studies minimax optimality over a class such as all shuffle-DP mechanisms. Can the authors discuss whether restricting attention to unbiased mechanisms is restrictive here, and to what degree it may exclude better biased procedures? This seems especially relevant in light of the bias-accuracy-privacy tradeoff highlighted in A Bias-Accuracy-Privacy Trilemma for Statistical Estimation (Kamath et al., 2023).

- The paper contains many definitions and technical constructions that are difficult to absorb. It would benefit from more intuition and explanation around the main definitions.

- The paper relies on results from Takagi & Liew (2026), which was only very recently uploaded to arXiv. While this is not an issue in itself, it does mean that some of the technical ingredients used here have not yet undergone peer review.

Additional Comments

- The sentence “The class $\mathcal R$ is substantially broader than the class of pure LDP mechanisms” is not exactly correct, since, as pointed out in Appendix B, trivial mechanisms for instance do not satisfy this property.

- It would be helpful if the authors could explicitly summarize which technical tools introduced in this work are new and may be of broader interest to the private ML community.

- The statement around line 177 (left column) seems to require either a justification or a pointer to a proof.

- The term “minimax” is usually used in a statistical setting, whereas the guarantees studied here seem closer to instance-optimality statements. I think a remark clarifying this terminology would be welcome.

---

> ### Author Rebuttal · Authors · 2026-03-31
>
> We thank the reviewer for the careful and thoughtful reading of our paper. We appreciate the detailed comments and questions, which were highly valuable to us. We plan to address all of them in the revision. We respond to the reviewer’s questions (Q), weaknesses (W), and additional comments (C).
> # Q1: Was the idea
> Yes. It appears in Balle et al. (2019) under the name privacy blanket. We follow Takagi & Liew in using the term blanket distribution.
> # Q2: Could the suboptimality
> We do not establish a suboptimality result for the standard Gaussian randomizer. In other words, we do not show that adding a blanket improves upon the Gaussian randomizer per se. Rather, the role of the blanket in BMG is analytical. It allows us to invoke the privacy blanket upper bound (Lemma 2.6), and obtain a provable privacy. Note that for the Gaussian randomizer without a blanket, the privacy-blanket upper bound is not effective. We can derive a utility in terms of $\chi_{\mathrm{chua}}$ for the Gaussian randomizer without a blanket (Section 4.3), but converting it into a DP guarantee depends on the conjecture (Chua et al. (2024)).
> # Q3: In Equation (1)
> Yes, this is because we work in a high-privacy scaling where $\varepsilon_n=O(\sqrt{\log n/n})$. A $1/n$-scale noise corresponds to the regime $\varepsilon_n\asymp \sqrt{\log(1/\delta_n)}$. In that regime, the privacy-utility trade-off is worse than the central setting (Asi et al., 2024). We identify the high-privacy regime (Definition 3.1) as the regime in which the shuffle model can approach the central model at the constant level.
> # Q4: If I understand correctly
> The intuition is that there are about $n\gamma$ messages on average coming from the common blanket component. Because the analyzer only sees the anonymized multiset of messages after shuffling, these blanket messages hide the target message: it becomes difficult to tell which message is attributable to the changed user in a neighboring dataset.
> # Q5: Is the unbiasedness
> The unbiasedness requirement is imposed on the aggregated estimator. Also, note that we induce the single-user unbiasedness requirment from the aggregated-level unbiasedness requirement via our reduction (Lemma 3.4).
> # Q6: In Proposition 4.2
> Thank you for pointing out this. e denotes an arbitrary unit vector. Because the Gaussian local randomizer is rotationally invariant, the quantity defining $\chi_{chua}$ does not depend on which unit vector is chosen.
> # Q7: Much of the analysis
> Our analysis does not take a simultaneous double limit. In Lemma 3.4, $\chi$ is treated as a fixed constant, and the asymptotics are only in $n\to\infty$. After this reduction, single-user optimization problems no longer depend on $n$. The asymptotics in which $\chi\to\infty$ are taken only at this local level (Corollary 3.6). Theorem 3.3 is stated for each fixed $\sigma>0$; the term $\eta(\sigma)=O(\sigma^{-2/3})$ only describes how the bound depends on $\sigma$, and the theorem itself does not claim a joint limit in $n$ and $\sigma$.
> # W1: The results seem, As stated by
> We refer to our response to Q1 from Reviewer M61n.
> # W2: The focus on unbiased
> In principle the unbiasedness constraint could exclude better biased procedures. However, we impose it because it enables a constant-level analysis (Theorem 3.3). Regarding Kamath et al. (2023), their trilemma is relevant, but their setting differs from ours. The trilemma is sharp in unbounded estimation problems where clipping is essential. Our focus is mean estimation over the bounded $\ell_2$ ball where clipping is not required, so that particular source of bias is absent. Even under unbiasedness, our protocol asymptotically matches the central Gaussian mechanism. To the best of our knowledge, we are not aware of prior work showing that allowing bias improves the leading-order rate over the Gaussian mechanism even in central DP (e.g., Chen et al. 2023); hence, we do not believe the leading-order rate improvement. We also note that this restriction is aligned with the leading mechanisms for LDP/shuffle vector mean estimation (Asi et al. (2022); Chen et al. (2023)), which are unbiased.
> # W3: The paper contains
> We refer to our responses to Q1 and W2 of reviewer M61n.
> # W4: The paper relies
> The paper appears to have been accepted to PODS 2026.
> # C1: The sentence
> You are right that the sentence is not accurate. We will revise the sentence to explicitly refer to pure LDP mechanisms with non-identical output distributions across at least one pair of inputs.
> # C2: It would be helpful
> We refer to our response to reviewer JSY7’s question on Theoretical Transparency.
> # C3: The statement around
> The statement around line 177 is justified by Takagi and Liew, Remark A.9, and we will add a pointer in the revision.
> # C4: The term “minimax”
> We agree that the term minimax is confusing, since our guarantees are not minimax in the usual statistical sense. Our results are stated in a dataset-level worst-case sense. We will stop using minimax.

---

> > ### Author Rebuttal · Reviewer_APgj · 2026-04-02
> >
> > I thank the authors for their rebuttal.

---

### Official Review · Reviewer_3Quu · 2026-03-13

**Soundness:** 4
**Presentation:** 3
**Significance:** 4
**Originality:** 3
**Overall Recommendation:** 5
**Confidence:** 3

**Summary:**

Companies like Apple and Google utilizes LDP and shuffle LDP to collect telemetry (e.g., about Genmoji or photo recommendation). However, while LDP is well understood, analysis of shuffle DP is much more involved since it is not local anymore and often LDP is analyzed and shuffle LDP result is obtained via amplification by shuffling.

This paper is trying bridge this gap and develop a theory for d-dimensional mean estimation in shuffle LDP regime.
The uses a scalar metric called shuffle index to create an optimization problem that allows constructing an optimal shuffle LDP mechanism. In addition, they proved that traditional LDP-optimal tools (like PrivUnit) fall short in high dimensions since their shuffle index is not tight.

Finally, the paper constructs a mechanism that is actually tight in the high privacy regime.

**Compliance With Llm Reviewing Policy:**

Affirmed.

**Key Questions For Authors:**

I don't have questions for the authors.

**Limitations:**

yes

**Strengths And Weaknesses:**

Soundness: The result in the paper are proven correctly.

Presentation: Submission is written clearly; however, it dives directly into the proofs without providing any intuition for the results which complicates reading the paper by people not actively working with shuffle LDP and LDP.

Significance: Mean estimation is one of the most widely used problems and shuffle LDP has a lot of significant application in practice. As a result having complete understanding of the problem is very important especially in the high privacy regime that is mostly desired.

Originality: The work provide new insights into the structure of shuffle LDP protocols.

---

> ### Author Rebuttal · Authors · 2026-03-31
>
> We thank the reviewer for the positive evaluation and encouraging comments. We also appreciate the suggestion regarding presentation, and we will work to improve the intuition and readability in the revision.

---

> > ### Author Rebuttal · Reviewer_3Quu · 2026-04-05
> >
> > Thank you for the response.

---

### Official Review · Reviewer_M61n · 2026-03-13

**Soundness:** 4
**Presentation:** 3
**Significance:** 3
**Originality:** 3
**Overall Recommendation:** 4
**Confidence:** 2

**Summary:**

This work studies the design of privacy algorithms for vector mean estimation under the single-message shuffle model. They study the choice of mechanism to optimize the accuracy under a shuffled DP constraint. They utilize the "shuffle index" from (Takagi & Liew, 2026) to characterize the privacy profiles of the shuffle mechanism. Built on the index, they formulate the post-shuffle mechanism design as an clean optimization problem. They then provide a minimax lower bounds for unbiased protocols, and construct an explicit mechanism that asymptotically matches the lower bound in the high privacy regime. They also validate the theoretical predictions via numerical experiments.

**Compliance With Llm Reviewing Policy:**

Affirmed.

**Final Justification:**

The response help me better understand the contribution of the paper compared to prior work. However, due to my low familiarity with the literature, I will keep my original score.

**Key Questions For Authors:**

Could you explain more regarding the scope of applicability of the high-privacy regime? When will this assumption be violated?

**Limitations:**

Yes.

**Strengths And Weaknesses:**

**Strengths**
- Soundness and presentation: The submission is technically sound. The results should be easy to read for readers who are familiar with related work and background. However, for non-experts, it would be nice to explain the intuition behind the definitions and approaches, especially the definition of the shuffle indices in Definition 2.5 and the definition of high-privacy regime in Definition 3.1.
- Originality and theoretical contribution: The paper nicely filled a gap existing in the previous literature. It is good to know that previous methods are suboptimal in this case, and the optimization problem formulation is clean. It is good to have a lower bound and an explicit mechanism that matches the lower bound (under some assumptions).

**Weaknesses**
- While the results are good, many of the main tools used to derive the results come from prior work, the problem formulation was introduced by Balle et al., 2019, the definition of the shuffle index came from Takagi & Liew, 2026. As a non-expert, it is a bit difficult for me to evaluate the significance of the incremental contribution.
- The assumption of the high-privacy regime should be better justified. The paper used the example of Gaussian mechanism to justify the high-privacy regime. But it is still difficult for me to understand the intuition and the scope of applicability for this assumption.

---

> ### Author Rebuttal · Authors · 2026-03-31
>
> We thank the reviewer for the constructive and positive review. We also appreciate the reviewer’s specific and valid comments on the weaknesses.  We will revise the paper to clarify these points, and we address them below.
>
> # Q1: Could you explain more regarding the scope of applicability of the high-privacy regime? When will this assumption be violated?
> To make the scope of the assumption more concrete, we can give a simple representative example of the $\sigma$-high privacy regime:
> $$
> \left(\varepsilon_n=\frac{1}{\sigma}\sqrt{\frac{\log n}{n}},\delta_n=\frac{1}{\sqrt{2\pi}\sigma n\log n}\right),
> $$
> $\delta_n$ scales as $O(1/n)$, which aligns with standard DP conventions. While $\varepsilon_n$ might appear restrictive, this regime is non-vacuous from a DP perspective: as discussed in Section 6, such high-privacy single-message primitives are the building blocks for multi-round shuffled protocols (Asi et al. (2024), Chen et al. (2023)), where composition can yield overall moderate privacy levels. Thus, rather than being an arbitrary technical restriction, this regime is of practical interest.
>
> $\sigma$ is a fixed constant, and our theory still applies for smaller $\sigma$, which corresponds to larger $\varepsilon_n$. However, as $\varepsilon_n$ becomes larger, the regime moves away from the window in which our lower and upper bounds are closest. In particular, Theorem 3.3 shows that the gap between the lower and upper bounds decreases as $\sigma$ increases, so the Gaussian-limit correspondence is sharpest for larger $\sigma$. This behavior is also reflected in Fig. 1. Thus, the main issue outside this regime is not that the theory becomes invalid, but rather that the privacy-utility trade-off is no longer guaranteed to be as close to the central Gaussian. In this sense, the assumption is mainly violated when $\varepsilon_n$ is taken substantially larger than $\sqrt{\log n/n}$.
>
> We will add a short explanation along these lines to clarify both the intuition and the practical scope of the assumption.
>
> # W1: While the results are good, many of the main tools used to derive the results come from prior work, the problem formulation was introduced by Balle et al., 2019, the definition of the shuffle index came from Takagi & Liew, 2026. As a non-expert, it is a bit difficult for me to evaluate the significance of the incremental contribution.
>
> We refer the reviewer to the response to Question (Theoretical Transparency) of reviewer JSY7, which explicitly summarizes our technical contributions.
>
> # W2: the intuition behind the definitions and approaches, especially the definition of the shuffle indices in Definition 2.5.
>
> Lemma 2.6 states that, in the high-privacy regime and for sufficiently large $n$, the upper bound and lower bound of privacy (profile) of the shuffled mechanism can be asymptotically characterized by scalar quantities that depend only on the local randomizer. These quantities are the lower and upper shuffle indices (Definition 2.5). This enables the conversion of the global shuffled $(\varepsilon_n,\delta_n)$-DP constraint into a scalar constraint in terms of shuffle indices.
>
> At an intuitive level, the shuffle index measures how distinguishable two local inputs (i.e., $x_1$ and $x_1’$) remain after passing through the local randomizer. Its definition is based on $\mathrm{Var}(\ell_0)$. Concretely, $\mathrm{Var}(\ell_0(Y;x_1, x_1’,\mathcal R_{\mathrm{ref}}))$ can be viewed as a $\chi^2$-type discrepancy of $R_{x_1}$ and $R_{x_1’}$ (from the view of $R_{\mathrm{ref}}$). A larger shuffle index means the locally randomized outputs are harder to distinguish, and stronger privacy is obtained after shuffling.
>
> The lower and upper shuffle indices play slightly different roles. The lower shuffle index governs the upper bound on the privacy profile, while the upper shuffle index governs the lower bound. In particular, the lower shuffle index is defined using the blanket distribution and blanket mass $\gamma$ because the upper-bound analysis relies on the privacy blanket technique of Balle et al. (2019).
> We will add this intuition near Definition 2.5 to clarify the meaning of the shuffle index.

---

> > ### Author Rebuttal · Reviewer_M61n · 2026-04-02
> >
> > Thank you for the response.

---

### Official Review · Reviewer_JSY7 · 2026-03-13

**Soundness:** 2
**Presentation:** 2
**Significance:** 3
**Originality:** 3
**Overall Recommendation:** 4
**Confidence:** 3

**Summary:**

This paper studies unbiased high-dimensional vector mean estimation in the single-message shuffle model, with the main goal of understanding how mechanism design should change after shuffling rather than simply inheriting designs that are optimal under local differential privacy. The paper introduces the notion of a **shuffle index** as the key analytical tool, reformulates post-shuffling mechanism design as a structured constrained optimization problem, derives a minimax lower bound centered around \( d\chi^2 \), shows that the classical PrivUnit mechanism may become suboptimal after shuffling, and further proposes a **blanket-mixed Gaussian (BMG)** mechanism that asymptotically matches the performance of the central Gaussian mechanism in the high-privacy regime. Overall, the paper addresses a well-motivated and technically meaningful question—namely, whether LDP-optimal mechanisms remain shuffle-optimal—and provides a coherent theoretical framework together with a concrete mechanism design that gives this question a compelling answer.

**Compliance With Llm Reviewing Policy:**

Affirmed.

**Key Questions For Authors:**

I encourage the authors to address the following points during the rebuttal or in the revised manuscript:

1. **Theoretical Transparency:** Could you clarify which components of the theoretical framework are derived entirely within this paper and which rely essentially on external asymptotic results regarding shuffled privacy profiles? A clearer distinction would improve the transparency of the theoretical contributions.
2. **Scope of Asymptotic Equivalence:** The current wording around the asymptotic equivalence result is somewhat ambiguous. Could you clarify whether the theorem is intended to apply specifically along a Gaussian-profile-matched sequence $(\varepsilon_n, \delta_n)$, or if it holds more broadly across a high-privacy region? Sharpening this definition would help avoid potential misinterpretation.
3. **Formula Verification (Section 4):** Please carefully verify the formulas in Section 4, specifically the substitution following:
   $$\chi_{\mathrm{lo}}^2 = \frac{\gamma}{(1-\gamma)^2}\cdot \frac{1}{e^{1/\sigma_0^2}-1}$$
   This relation appears to imply:
   $$\sigma_0^2 = \left[ \log\left( 1+\frac{\gamma}{(1-\gamma)^2\chi^2} \right) \right]^{-1}$$
   Consequently, the reciprocal of the logarithmic term should be preserved in the final error expression. Could you confirm if the current multiplicative form in the draft is a typographical error?
4. **Experimental Baselines:** To strengthen the empirical results, would it be possible to include matched-privacy comparisons against existing shuffled baselines, such as **shuffled PrivUnit**? Demonstrating improvements over established shuffled mechanisms would provide more compelling evidence for the BMG mechanism's utility than comparisons to the central Gaussian mechanism alone.

*How these answers affect my evaluation:* A clear resolution of the formula-level consistency (Q3) and a more explicit mapping of theoretical dependencies (Q1) would significantly increase my confidence in the technical soundness of the paper.

**Limitations:**

The limitations of the paper are reasonably clear:

* **Asymptotic Focus:** The strongest theoretical results are asymptotic and operate in a high-privacy regime. As a result, the work provides an **asymptotic near-optimal characterization** rather than a fully sharp characterization for all $(\varepsilon, \delta)$ parameters across all regimes.
* **Assumption of Unbiasedness:** The analysis focuses primarily on unbiased estimators. While this assumption simplifies the minimax analysis and improves interpretability, it may exclude biased estimators that could potentially achieve better privacy–utility tradeoffs in practical scenarios. The authors acknowledge this possibility, which is an appropriate and transparent limitation.
* **Scope of Resolution:** Overall, the work should be viewed as a significant theoretical step toward understanding post-shuffling optimal estimation, rather than the final resolution of the problem.

These constraints do not diminish the paper's value but suggest that future work could explore non-asymptotic bounds and the role of biased estimation in the shuffle model.

**Strengths And Weaknesses:**

**Strengths:**

* **Important Problem:** The gap between local privacy design and shuffle-aware design is both conceptually interesting and practically relevant, and the paper identifies this gap very clearly.
* **Novel Viewpoint:** The shuffle-index viewpoint is novel and useful, as it turns a difficult post-shuffling privacy analysis into a more explicit optimization framework that supports lower bounds, upper bounds, and asymptotic comparisons within a unified language.
* **Constructive Mechanism:** The BMG mechanism is simple, interpretable, and theoretically well-motivated, providing a constructive answer rather than only a lower-bound or impossibility result.

**Areas for Revision:**
* **External Dependencies:** Some of the strongest conclusions rely essentially on recent external asymptotic results. The manuscript could be more explicit about which claims are fully self-contained and which depend on those external ingredients.
* **Scope Clarification:** The main asymptotic equivalence result is established along a specific high-privacy scaling of $(\varepsilon_n, \delta_n)$, but the current wording may lead some readers to interpret it as a broader region-wise statement.
* **Formula Verification:** Section 4 would benefit from a careful recheck of the displayed formulas. Starting from:
$$\chi_{\mathrm{lo}}^2 = \frac{\gamma}{(1-\gamma)^2}\cdot \frac{1}{e^{1/\sigma_0^2}-1}$$
one obtains:
$$\sigma_0^2 = \left[\log\left(1+\frac{\gamma}{(1-\gamma)^2\chi^2}\right)\right]^{-1}$$
Therefore, the corresponding risk expression should preserve the reciprocal of the logarithmic term.
* **Empirical Validation:** The experimental section illustrates the theoretical trend, but it would be stronger with more direct matched-privacy comparisons against natural shuffled baselines such as PrivUnit or scalarized alternatives.

*(Note: These issues seem to be revision points rather than reasons to reject the paper.)*

---
**Soundness**

The paper is technically substantial and the overall proof strategy is sensible: the main theorem sequence, the lower-bound argument, the asymptotic correspondence, and the constructive mechanism design fit together in a logically meaningful way. In that sense, the work is clearly nontrivial and built on serious analysis. That said, the current version is not yet at the level of complete technical transparency.

* **Logical Dependence:** Some central claims depend on a recent external asymptotic characterization of shuffled privacy profiles. The manuscript could clarify more explicitly the logical role of that dependence.
* **Formula-Level Issue:** In the BMG derivation, if:
$$\sigma_0^2 = \left[\log\left(1+\frac{\gamma}{(1-\gamma)^2\chi^2}\right)\right]^{-1}$$
then the corresponding single-user error should be written as:
$$\mathrm{Err}_1 = \frac{d}{(1-\gamma)^2} \left[\log\left(1+\frac{\gamma}{(1-\gamma)^2\chi^2}\right)\right]^{-1} + \frac{\gamma}{1-\gamma}$$
rather than with the logarithm appearing multiplicatively. This may simply be a typographical issue, but since later asymptotic expansions rely on this expression, it deserves careful verification.

---

**Presentation**

The paper is generally well written and organized, especially in terms of motivation, high-level contribution statements, and the progression from problem formulation to mechanism construction. The introduction effectively explains why shuffle-aware mechanism design is distinct from standard LDP mechanism design. However, the presentation could be improved in several ways:

* **Scope of Asymptotic Statement:** This should be described more carefully so readers do not confuse sequence-wise asymptotic correspondence with a uniform characterization over an entire privacy region.
* **Roadmap for Technical Quantities:** While $\chi_{\mathrm{lo}}$, $\chi_{\mathrm{up}}$, and $\chi_{\mathrm{chua}}$ are meaningful in context, the manuscript would benefit from a clearer “roadmap” paragraph explaining their roles in the lower bound, upper bound, and conjecture-based discussion.
* **Manuscript Quality:** Minor quality issues should be cleaned up during revision to noticeably improve readability.

---

**Significance**

The significance of this work is solid. The single-message shuffle model is an important intermediate privacy model between local and central differential privacy, and high-dimensional mean estimation is one of the most representative statistical tasks in this area. More importantly, the paper reframes the problem through the lens of **shuffle-aware mechanism design** and shows that mechanisms optimal under LDP are not necessarily optimal once shuffling is introduced. This conceptual distinction is important and may influence future research.

---

**Originality**

The paper demonstrates originality in both viewpoint and technical development. Previous work has often focused on privacy amplification or the shuffled behavior of mechanisms originally designed for the local model. In contrast, this paper explicitly investigates how mechanisms should be designed **after shuffling is taken into account as a first-class design constraint**. The introduction of the shuffle index provides a useful conceptual tool, applied effectively to prove the suboptimality of PrivUnit and motivate the BMG mechanism.

---

> ### Author Rebuttal · Authors · 2026-03-31
>
> We thank the reviewer for the careful reading and for the constructive and helpful comments. Below, we respond to the key questions in turn.
>
> # Question (Theoretical Transparency)
>
> Concretely, our main result (Theorem 3.3) has two conceptually distinct layers.
> 1. External ingredient used essentially.
> The step that converts the global shuffled $(\varepsilon_n,\delta_n)$-DP constraint into a constraint in terms of shuffle indices relies essentially on the Lemma 2.6 (Takagi & Liew (2026)). The high privacy regime (Definition 3.1) is chosen precisely to match the asymptotic form in Lemma 2.6.
>
> 2. New contributions of this paper.
> Conditional on that reduction, the subsequent optimization analysis is developed in this paper. This includes:
>  - the reduction from the $n$-user unbiased shuffled setting to the single-user optimization problem with shuffle indices (Lemma 3.4, building on reduction ideas from Asi et al. (2022) in the Local DP setting but adapted here to the shuffle-index setting);
>  - the lower bound (Theorem 3.5);
>  - the constructive upper bound via the blanket-mixed Gaussian mechanism (Corollary 3.6 and Section 4); and
>  - the suboptimality result for PrivUnit (Proposition 3.7).
>
> We acknowledge that the paragraph on p.1 (line 036 in the right column, beginning “We address this gap using the recently proposed shuffle index...”) may blur the distinction between the imported ingredient from Takagi & Liew (2026) and our new contributions. We will revise the introduction to make the above theoretical contributions explicit.
>
> # Question (Scope of Asymptotic Equivalence)
>
> The definition of the $\sigma$-high privacy regime is parameterized by a Gaussian-profile-matched sequence for a fixed constant $\sigma>0$. This is used to calibrate the privacy level.
>
> At the same time, the actual privacy requirement in the optimization problem is one-sided, as shown in Eq. (4): $\delta_{\mathcal S\circ \mathcal R^n}(\varepsilon_n)\le \delta_n.$
> Therefore, the feasible class is broader than mechanisms whose privacy profile exactly matches the Gaussian benchmark: it also includes any mechanism achieving asymptotically smaller $\delta_n$, i.e., stronger privacy at the same $\varepsilon_n$. In this sense, the matched sequence should be viewed as a benchmark level used to parameterize the regime, while the corresponding privacy constraint covers a broader high-privacy region below that benchmark. We will revise the wording to make this distinction explicit.
>
> Also, we refer you to the response to Q1 of Reviewer M61n for the intuition of the high-privacy regime.
>
> # Question (Formula Verification (Section 4))
>
>
> We carefully rechecked the formula in Section 4. In the submitted manuscript, the logarithmic term already appears in the denominator:
>
> $$
> \mathrm{Err}_1= \frac{d}{(1-\gamma)^2\log \left(1+\frac{\gamma}{(1-\gamma)^2\chi^2}\right)} +\frac{\gamma}{1-\gamma}.
> $$
>
> This is consistent with the substitution implied by
>
> $$
> \chi_{\mathrm{lo}}^2=\frac{\gamma}{(1-\gamma)^2}\cdot \frac{1}{e^{1/\sigma_0^2}-1},
> $$
>
> namely
>
> $$
> \sigma_0^2= \left[ \log \left(1+\frac{\gamma}{(1-\gamma)^2\chi^2}\right) \right]^{-1}.
> $$
>
> So there is no intended multiplicative-log form in the submitted version. If we may be misunderstanding the reviewer’s concern, we would be grateful for any further clarification and would be happy to recheck the relevant passage accordingly.
>
> # Question (Experimental Baselines)
>
> We have added the comparison against shuffled PrivUnit in Figure 1. You can see the result here: https://drive.google.com/file/d/1lkZEgOUQngUyfZkQM3Xb7cKGc-daNuJQ/view?usp=sharing. The new experiment shows that, consistently with Proposition 3.7, the utility gap between shuffled PrivUnit and the central Gaussian does not vanish: as privacy becomes stronger, shuffled PrivUnit continues to exhibit a constant-factor gap, empirically matching the (\sqrt{\pi/2}) predicted by Proposition 3.7. In contrast, BMG attains the central Gaussian score much more closely, which empirically illustrates the distinction between the suboptimality of PrivUnit and the asymptotic optimality of BMG.

---

### Decision · Program_Chairs · 2026-04-30

**Decision:**

Accept (regular)

**Comment:**

This paper makes a solid contribution to the theory of the shuffle model of DP. The paper argues that optimal mechanism design for shuffle-DP is different from optimal design of LDP mechanisms. In particular, it shows that traditional LDP-optimal tools such as PrivUnit, when used for shuffle-DP, can fall short in high dimensions, and it provides an alternative mechanism. Reviewers agreed that the direction is interesting and that the shuffle-index viewpoint is novel and useful.

The rebuttal addressed several concerns, including clarifying which parts of the proofs are new versus borrowed from prior work, which privacy regime the results apply to, and adding a baseline comparison to shuffled PrivUnit. These clarifications should be incorporated into the final version. Overall, the rebuttal helped support the view that the paper makes a meaningful theoretical contribution.

That said, the paper is technically dense, and I found the presentation difficult to follow in places. In particular, the use of asymptotics and the precise scope of the high-privacy regime could be explained more clearly. Because of this, I am not fully confident in my ability to assess all details of the correctness. Several reviewers also raised limitations around the asymptotic nature of the main results, the focus on the high-privacy regime, and the restriction to unbiased estimators.

I therefore recommend weak acceptance.